# A universal probabilistic spike count model reveals ongoing modulation of neural variability

**David Liu**
Department of Engineering
University of Cambridge
dl543@cam.ac.uk

**Máté Lengyel**
Department of Engineering       Department of Cognitive Science
University of Cambridge         Central European University
m.lengyel@eng.cam.ac.uk

## Abstract

Neural responses are variable: even under identical experimental conditions, single neuron and population responses typically differ from trial to trial and across time. Recent work has demonstrated that this variability has predictable structure, can be modulated by sensory input and behaviour, and bears critical signatures of the underlying network dynamics and computations. However, current methods for characterising neural variability are primarily geared towards sensory coding in the laboratory: they require trials with repeatable experimental stimuli and behavioural covariates. In addition, they make strong assumptions about the parametric form of variability, rely on assumption-free but data-inefficient histogram-based approaches, or are altogether ill-suited for capturing variability modulation by covariates. Here we present a universal probabilistic spike count model that eliminates these shortcomings. Our method builds on sparse Gaussian processes and can model arbitrary spike count distributions (SCDs) with flexible dependence on observed as well as latent covariates, using scalable variational inference to jointly infer the covariate-to-SCD mappings and latent trajectories in a data efficient way. Without requiring repeatable trials, it can flexibly capture covariate-dependent joint SCDs, and provide interpretable latent causes underlying the statistical dependencies between neurons. We apply the model to recordings from a canonical non-sensory neural population: head direction cells in the mouse. We find that variability in these cells defies a simple parametric relationship with mean spike count as assumed in standard models, its modulation by external covariates can be comparably strong to that of the mean firing rate, and slow low-dimensional latent factors explain away neural correlations. Our approach paves the way to understanding the mechanisms and computations underlying neural variability under naturalistic conditions, beyond the realm of sensory coding with repeatable stimuli.

## 1 Introduction

Classical analyses of neural coding are based on mean spike counts or neural firing rates. Indeed, some of the most paradigmatic examples of the neural code were discovered by regressing neural firing rates to particular sensory stimuli [1, 2] or behavioural covariates [3, 4, 5, 6] to characterize their tuning properties. However, neural spiking is generally not regular. Recordings from many cortical areas show significantly different activity patterns within and across identical trials [7], despite fixing experimentally controlled variables. This irregularity is also seen in continual neural recordings without trial structure [8]. The resulting variability has classically been characterised as 'Poisson', with a Fano factor (variance to mean ratio) of one [9], but experimental data also often exhibits significantly more [10, 8, 11, 12] and sometimes less [13, 14] variability, respectively referred to as over- or underdispersion. Moreover, experimental studies have revealed that neural variability generally depends on stimulus input and behaviour [15, 16, 17, 18], and exhibits structured shared

35th Conference on Neural Information Processing Systems (NeurIPS 2021).

variability ('noise correlations') across neurons even after conditioning on such covariates. Such correlations can have important consequences for decoding information from neural population activity [19, 20, 21] and reveal key properties of the underlying circuit dynamics [22]. Moreover, theories of neural representations of uncertainty have assigned computational significance to variability as a signature of Bayesian inference [23, 24, 25, 26]. Thus, just as classical tuning curves for firing rates have been crucial for understanding some of the fundamental properties of the neural code, a principled statistical characterisation of neural variability, and its dependence on stimulus and behavioral covariates, is a key step towards understanding the dynamics of neural circuits and the computations they subserve.

The traditional approach to characterising neural variability has been pioneered in sensory areas, and relies on repeatable trial structure with a sufficiently large number of trials using identical stimulus and behavioral correlates [27, 15, 28]. Variability in this case can be quantified by simple summary statistics of spike counts across trials of the same condition. However, this approach does not readily generalise to more naturalistic conditions where covariates cannot be precisely controlled and repeated in an experiment. This more general setting requires statistical methods that take into account temporal variation of covariates for predicting neural count activity. Generalised Linear Models are a popular choice [29], but they only model the dependence of firing rates on covariates – with changes in variability directly coupled to changes in the rate inherent to Poisson spiking. More complex methods for inferring neural tuning [30, 31] and latent structure [32, 33, 34, 35] similarly use restrictive parametric families for spike count distributions, and thus also cannot model changes in variability that are not 'just' a consequence of changes in mean counts or firing rates. Conversely, statistical models capable of capturing arbitrary single neuron count statistics, such as histogram-based approaches or copulas [36], do not incorporate dependencies on covariates.

Here we unify these separate approaches, resulting in a single framework for jointly inferring neural tuning, single neuron count statistics, neural correlations, and latent structure. Our semi-parametric approach leads to the universal count model (UCM) for counts ranging from 0 to $K$, in the sense that we can model arbitrary distributions over the joint count space of size $(K+1)^N$ of $N$ neurons. The trade-off between computational overhead and model expressivity is controlled by hyperparameters, with expressivity upper bounded by the true universal model. Our approach extends the idea of a universal binary count model [37] to a finite range of integer counts, while allowing flexible dependence on observed and latent covariates to model non-stationary neural activity and correlations. The flexibility reduces biases from restrictive assumptions in any of the model components. Scalability is maintained by leveraging sparse Gaussian processes [38] with mini-batching [39, 40] to handle the size of modern neural recordings.

We first define the UCM, and then describe how to interpret as well as evaluate model fits. As our model is able to capture arbitrary single neuron statistics, we build on the Kolmogorov-Smirnov test to construct more absolute goodness-of-fit measures. After validating our method on synthetic data that cannot be captured by currently used methods, we apply the model to electrophysiological recordings from two distinct brain regions in mice that show significant tuning to the head direction of the animal [41, 42]. We find that (1) neural activity tends to be less dispersed than common Poisson-like models at higher firing rates, and more dispersed at low rates; (2) mean and variance of counts defy a simple parametric relationship imposed by parametric count distribution families; (3) variability modulation by behaviour can be comparable or even exceed that of the mean count or firing rate; (4) a two-dimensional latent trajectory varying on timescales of $\sim 1$ s is sufficient to explain away neural correlations but not the non-Poisson nature of single neuron variability. Finally, we discuss related work, limitations and proposed extensions of our model.

## 2  Universal count model

**Notation**   Spike count activity of $N$ neurons recorded into $T$ time bins is formally represented as an $N$-dimensional time series of non-negative integers. Due to biological constraints, the possible spike counts have some finite upper bound $K$, taken as the highest observed count. We denote probabilities of a spike count distribution (SCD) by a vector $\boldsymbol{\pi}$ of length $K + 1$, and use $\Pi$ to denote the collection of vectors $\boldsymbol{\pi}_{nt}$ for neurons $n$ and time steps $t$. Additionally, we denote the count activity by a matrix $Y \in [0, K]^{N \times T}$ with elements $y_{nt}$. Input covariates are observed $X \in \mathbb{R}^{T \times D_x}$ (e.g. animal speed) or latent $Z \in \mathbb{R}^{T \times D_z}$ (to capture e.g. attention), with range depending on topology [43]. We denote their elements $x_{td}$ and $z_{tq}$ with observed and latent dimension $d$ and $q$, respectively.

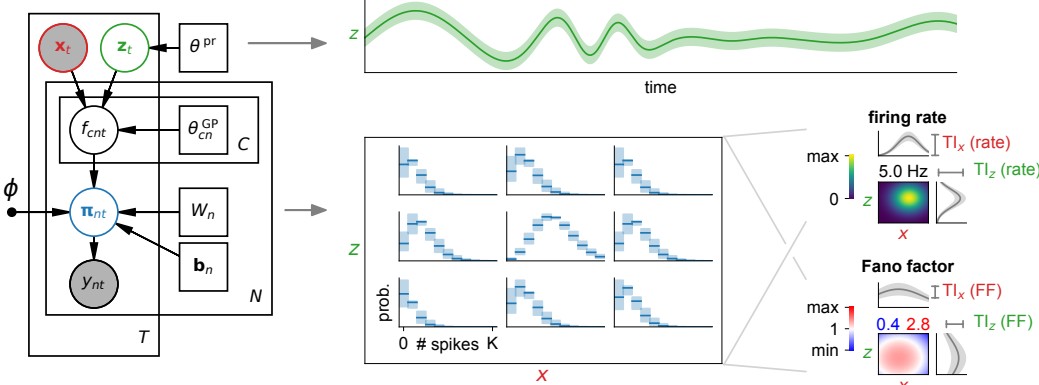

Figure 1: **Schematic of the UCM and the workflow.** Left: graphical model corresponding to Equation 1, with shaded circles as observed, open circles as latent, and squares as deterministic variables. Filled dots represent fixed quantities. Middle: example inference of model posterior Equation 3, with inferred latent trajectories (green, top) and covariate-dependent SCDs (blue, bottom) that depend on both observed $x$ and latent $z$ covariates. Note we only show the posterior over a single SCD evaluated on a $(x,z)$ grid, whereas the full posterior defines SCDs over all neurons. Right: obtaining interpretable spike count statistics from the SCDs (see subsection 2.3). Examples show firing rate and Fano factor tuning curves over observed $x$ and latent $z$ covariates, either jointly (heatmaps) or marginalized (grey curves). The depth of modulation in marginalized tuning curves is used to extract a tuning index (TI) for the chosen subsets of covariates, see Equation 6.

## 2.1 Generative model

The big picture is to model counts $Y$ with dependence on $X$. For each neuron, our model consists of $C$ Gaussian process (GP) priors, a basis expansion $\boldsymbol{\phi} : \mathbb{R}^C \to \mathbb{R}^{\tilde{C}}$, and a linear-softmax mapping

$$
\begin{aligned}
z_{tq} &\sim p(Z; \theta^{\mathrm{pr}}), \quad h_{cn}(\cdot) \sim \mathcal{GP}(0, k_{cn}(\cdot, \cdot; \theta_{cn}^{\mathrm{GP}})) \\
f_{cnt} &= h_{cn}(\boldsymbol{x}_t, \boldsymbol{z}_t) \\
\boldsymbol{\pi}_{nt} &= \mathrm{softmax}(W_n \, \boldsymbol{\phi}(\boldsymbol{f}_{nt}) + \boldsymbol{b}_n) \\
y_{nt} &\sim \mathrm{Discrete}(\boldsymbol{\pi}_{nt})
\end{aligned}
\tag{1}
$$

where $W \in \mathbb{R}^{(K+1) \times \tilde{C}}$, $\boldsymbol{b} \in \mathbb{R}^{K+1}$, and the GP covariance functions $k_{cn}$ have hyperparameters $\theta_{cn}^{\mathrm{GP}}$. The use of non-parametric GP mappings with point estimates for $W$ and $\boldsymbol{b}$ leads to a semi-parametric model with parameters $\theta$, see details in Appendix E. The overall generative model $P_\theta(Y|X)$ is depicted schematically in Figure 1. Note the model specifies a prior $p(\Pi|X)$ over joint SCDs, conceptually similar to Dirichlet priors [37] but allowing non-parametric dependence on $X$. With latent input $Z$, our model can flexibly describe multivariate dependencies in joint SCDs as conditional independence across neurons no longer holds when marginalizing over $Z$ [44]. In addition, $p(Z)$ models temporal correlations in the latent states. We use Markovian priors (details in subsection E.2)

$$
p_\theta(Z) = p_\theta(\boldsymbol{z}_1) \prod_{t=2}^{T} p_\theta(\boldsymbol{z}_t | \boldsymbol{z}_{t-1})
\tag{2}
$$

with $\theta^{\mathrm{pr}}$ absorbed into model parameters $\theta$ for compactness. This allows the model to flexibly capture both neural and temporal correlations in $Y$. To attain scalability, we use sparse GPs [38].

Depending on $C$ and basis functions $\phi(\cdot)$, we obtain an approximation to the true universal prior on joint SCDs, with 'universal' referring to the ability to capture any joint SCD over all neurons. Arbitrary single neuron statistics can be captured when $C = K$ with $\boldsymbol{\phi}(\boldsymbol{f}) = \boldsymbol{f}$, but fitting is computationally expensive when $N \times C \gg 1$. For capturing all correlations, the model also requires a sufficiently large latent space. One controls the trade-off between model expressiveness and computational overhead through $C$ and $\phi$. Larger expansions $\phi$ allow one to model count distributions more expressively with small $C$, e.g. the element-wise linear-exponential $\boldsymbol{\phi}(\boldsymbol{f}) = (f_1, e^{f_1}, \ldots, f_C, e^{f_C})$ covers a range of distributions including the truncated Poisson with only $C = 1$ (see subsection A.3).

## 2.2 Stochastic variational inference and learning

For the joint model distribution $p_\theta(Y, \Pi, Z|X) = P(Y|\Pi) \, p_\theta(\Pi|X, Z) \, p_\theta(Z)$, with count distributions $P(Y|\Pi)$, we approximate the posterior by $q_{\theta,\chi,\varphi}(\Pi, Z|X)$ that factorizes in the form

$$q_{\theta,\chi}(\Pi|X, Z) \, q_\varphi(Z) = \left( \prod_n^N q_{\theta,\chi}(\Pi_n|X, Z) \right) \left( \prod_t^T q_\varphi(\boldsymbol{z}_t) \right) \tag{3}$$

with $\varphi$ and $\chi$ the variational parameters for latent states and the sparse Gaussian process posterior (Appendix E), respectively. Note that we use a factorized normal $q(Z)$ for Euclidean $Z$, and a wrapped normal for circular $Z$ based on the framework of reparameterized Lie groups [45, 43]. The posterior over count probabilities $q_{\theta,\chi}(\Pi|X, Z)$ is defined as mapping the sparse Gaussian process posterior $q_{\theta,\chi}(F|X, Z)$ through $\Pi(F)$ (Equation 1), a deterministic mapping. This is analytically intractable, so in practice it is represented by Monte Carlo samples. Using stochastic variational inference [46], we minimize an upper bound on the negative log marginal likelihood

$$\mathcal{F}_{\theta,\chi,\varphi}(Y|X) = -\mathbb{E}_{Z \sim q_\varphi(Z)} \mathbb{E}_{\Pi \sim q_{\theta,\chi}(\Pi|X, Z)} \left[ \log \frac{P(Y|\Pi) \, p_\theta(\Pi|X, Z) \, p_\theta(Z)}{q_{\theta,\chi}(\Pi|X, Z) \, q_\varphi(Z)} \right] \tag{4}$$

known as the variational free energy. This objective leads to tractable terms (subsection E.1), allowing us to infer the approximate posterior as well as a lower bound of the log marginal likelihood [47, 40]. We use Adam [48] for optimization, see details of implementation and model fitting in Appendix E.

## 2.3 Obtaining interpretable spike count statistics from the model

**Characterizing spike count distributions** From the posterior $q(\Pi|X)$[1], we can compute samples of the posterior of any statistic of spike counts as a function of covariates. Single neuron statistics in particular can be characterized by tuning curves for both mean firing rates and Fano factors (FF)

$$\rho(X) = \frac{1}{\Delta} \mathbb{E}_{q(\Pi|X)} \mathbb{E}_{P(Y|\Pi)}[Y] \qquad \text{FF}(X) = \mathbb{E}_{q(\Pi|X)} \left[ \frac{\text{Var}_{P(Y|\Pi)}[Y]}{\mathbb{E}_{P(Y|\Pi)}[Y]} \right] \tag{5}$$

with time bin length $\Delta$. The model also quantifies private neuron variability that cannot be explained away by regressing to shared input (both observed and latent) through $P_n(y_{tn}|\boldsymbol{x}_t)$.

To quantify the sensitivity of a some aspect of neuron activity to a set of covariates $\boldsymbol{x}_*$, we define a tuning index (TI) with respect to a count statistic $T_y(\boldsymbol{x}_*)$

$$\text{TI} = \frac{\max_{\boldsymbol{x}_*} T_y(\boldsymbol{x}_*) - \min_{\boldsymbol{x}_*} T_y(\boldsymbol{x}_*)}{\max_{\boldsymbol{x}_*} T_y(\boldsymbol{x}_*) + \min_{\boldsymbol{x}_*} T_y(\boldsymbol{x}_*)} \tag{6}$$

with $T_y(\boldsymbol{x}_*)$ evaluated under the mean posterior SCD marginalized over all other covariate dimensions complementary to $\boldsymbol{x}_*$. These marginalized distributions are computed using observed input $\boldsymbol{x}_t$ (subsection D.5). Resulting marginalized tuning curves for TIs are depicted conceptually in Figure 1.

**Generalized $Z$-scores and noise correlations** The deviation of activity from the predicted statistics is commonly quantified through $Z$-scores [8, 49, 17], which are computed as $(y - \langle y \rangle)/\sqrt{\langle y \rangle}$ with $\langle y \rangle$ being the mean count in some time bin. If neural activity follows a Poisson distribution, the distribution of $Z$-scores asymptotically tends to a unit normal when average counts $\langle y \rangle \gg 1$ (Appendix B). To generalize the normality of the $Z$-score for arbitrary counts and SCDs, we use

$$\xi = \Phi^{-1}(u) \quad \text{with} \quad u(y) = \int_0^{y+\epsilon} p(\tilde{y}) \, \mathrm{d}\tilde{y} = \sum_{k=0}^{y-1} P(k) + \epsilon P(y), \quad \epsilon \sim \mathcal{U}(0, 1) \tag{7}$$

with $\Phi(\cdot)$ the unit Gaussian cdf., and dequantization noise $\epsilon$ to get continuous quantiles $u \sim \mathcal{U}(0, 1)$ and generalized $Z$-scores $\xi \sim \mathcal{N}(0, 1)$ from the probability integral transform.

With $\xi$, one can completely describe single neuron statistics with respect to the model. Correlations in the neural activity however will cause $\xi$ to be correlated. We define generalized lagged correlations $r_{ij}(\Delta) \in [-1, 1]$ and Fisher $Z \in \mathbb{R}$ that is more convenient for statistical testing

$$r_{ij}(\Delta) = \langle \xi_i(t) \, \xi_j(t + \Delta) \rangle_t, \quad Z_{\text{Fisher}} = \frac{1}{2} \log \frac{1 + r}{1 - r} \tag{8}$$

which describes spatio-temporal correlations not captured by the model. Noise correlations [50] refer to the case of $\Delta = 0$, when $r_{ij}$ becomes symmetric.

---

[1]For notational convenience, $X$ denotes both observed and latent covariates here.

## 2.4 Assessing model fit

Our model depends on a hyperparameter $C \leq K$ that trades off flexibility with computational burden. In practice, one likely captures the neural activity accurately with $C$ well below $K$ and a simple basis expansion as the linear-exponential above or quadratic-exponential $\phi(\boldsymbol{f}) = (f_1, f_1^2, e^{f_1}, \ldots, f_1 f_2, \ldots)$. This can be quantified by the statistical measures provided below, and allows us to select appropriate hyperparameters to capture the data sufficiently well.

To assess the model fit to neural spike count data, a conventional machine learning approach is to evaluate the expected log-likelihood of the posterior predictive distribution on held-out data $Y$, leading to the cross-validated log-likelihood

$$\text{cvLL} = \mathbb{E}_{q(Z)} \mathbb{E}_{q(\Pi|X,Z)} [\log P(Y|\Pi)] \tag{9}$$

where we cross-validate over the neuron dimension by using the majority of neurons to infer the latent states $q(Z)$ in the held-out segment of the data, and then evaluate Equation 9 over the remaining neurons. Without latent variables, we simply take the expectation with respect to $q(\Pi|X)$. However, the cvLL does not reveal how well the data is described by the model in an absolute sense. Likelihood bootstrap methods are possible [28], but become cumbersome for large datasets. To assess whether the neural data is statistically distinguishable from the single neuron statistics predicted by the model, we use the Kolmogorov-Smirnov framework [51] with $u$ from Equation 7 across time steps $t$

$$T_{\text{KS}} = \max_t |F_T(u_t) - u_t| \tag{10}$$

with empirical distribution function $F_T(u)$ (for details see Appendix B). This scalar number is positive and does not indicate whether the data is under- or overdispersed relative to the model. For this, a useful measure of dispersion is the logarithm of the variance of $\xi$ with a correction

$$T_{\text{DS}} = \log \langle \xi_t^2 \rangle_t + \left( \frac{1}{T} + \frac{1}{3T^2} \right) \tag{11}$$

which provides a real number with positive and negative sign indicating over- and underdispersion, respectively. Its sampling distribution under $\xi \sim \mathcal{N}(0, 1)$ is asymptotically normal, centered around 0 (due to the additive parenthetical term) with a variance depending on the number of timesteps $T$ (subsection B.3). This extends the notion of over- and underdispersion beyond the usual definition of dispersion relative to Poisson models [52]. To quantify whether the model has captured noise correlations in the data, we compute $\xi$ with respect to the mean posterior predictive distribution

$$Q_{\theta,\varphi}(Y|X) = \int \prod_t^T \left( \prod_n^N \mathbb{E}_{q(\boldsymbol{\pi}_{nt}|\boldsymbol{x}_t, \boldsymbol{z}_t)} [P(y_{tn}|\boldsymbol{\pi}_{nt})] \right) q_\varphi(\boldsymbol{z}_t) \, \mathrm{d}\boldsymbol{z}_t \tag{12}$$

which whitens $\xi$, hence reducing noise correlations $r$ in Equation 8, if correlations are explained away by co-modulation of neurons due to shared low-dimensional factors [44]. These are captured through latent states $Z$ using the posterior $q_\varphi(Z)$, inferred from the same data used to compute $\xi$. Overall, this can be interpreted as treating $Z$ as if it was part of the observed input to the model.

## 3 Results

In the following, we use $C = 3$ with an element-wise linear-exponential basis expansion (subsection 2.1). This empirically provided sufficient model flexibility as seen in goodness-of-fit metrics. We use an RBF kernel with cosine distances in case of angular input dimensions (subsection E.3).

### 3.1 Synthetic data

Animals maintain an internal estimate of their head direction [4, 42, 53]. Here, we extend simple statistical models of head direction cell populations [54] for validating the ability of the UCM to capture complex count statistics, as well as neural correlations through latent structure. The task is to jointly recover the ground truth count likelihoods, their tuning to covariates, and latent trajectories if relevant from activity generated using two synthetic populations. The first population was generated with a parametric heteroscedastic Conway-Maxwell-Poisson (CMP) model [55], which has decoupled mean and variance modulation as well as simultaneously over- and underdispersed activity (Fano

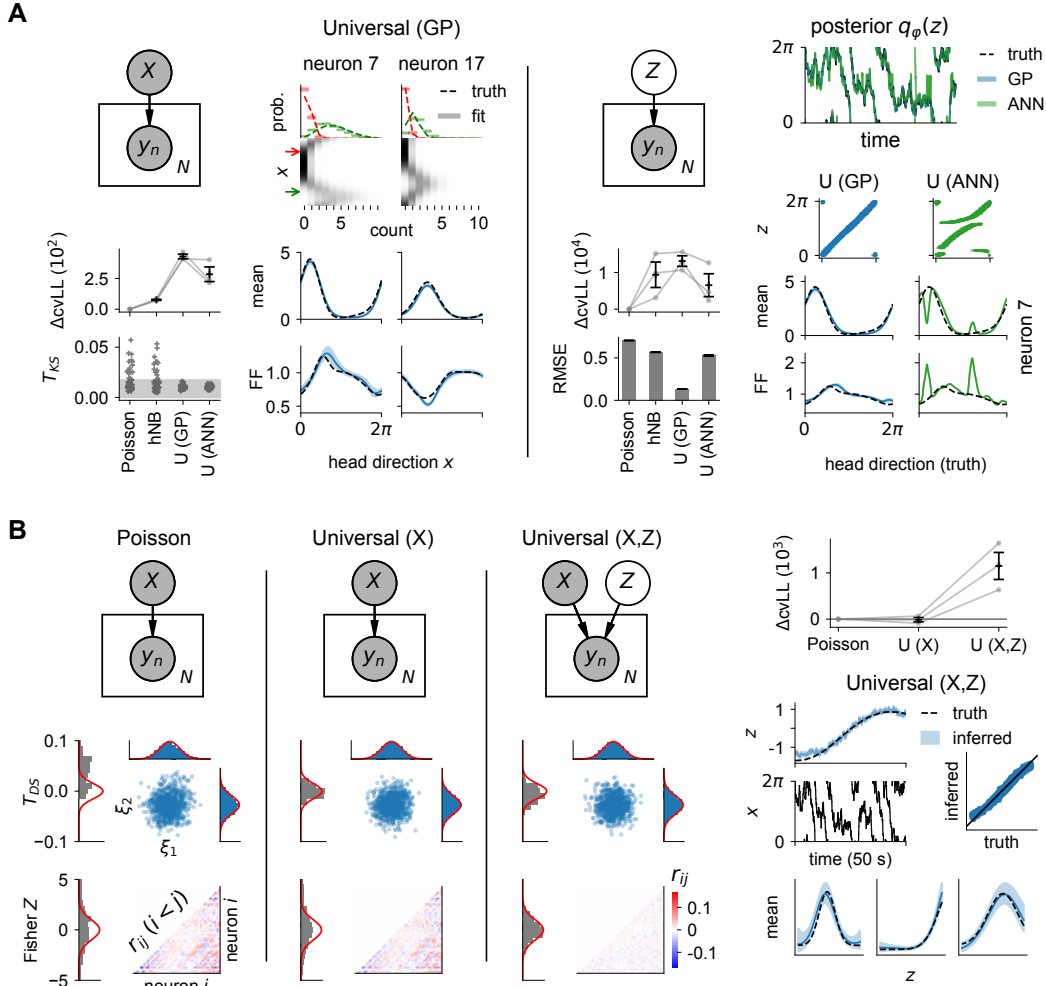

Figure 2: **Model validation with two synthetic head direction cell populations.** **(A)** Applying Poisson, heteroscedastic negative binomial (hNB) and universal (U, with either GP or ANN mappings) models to synthetic data from the heteroscedastic CMP population. Error bars indicate s.e.m. over cross-validation runs. Shaded regions for tuning curves and $T_{\text{KS}}$ indicate the $95\%$ CI. $\Delta$cvLL is the difference w.r.t. Poisson baseline. Left: regression models, visualizing SCDs (top right), tuning curves (bottom right), and fitting scores (bottom left) for two representative cells. Right: models with an angular latent variable, visualizing inferred latent states (top right), comparing the GP to ANN model (bottom right), and plotting fitting scores (bottom left). Root-mean-squared errors (RMSE) between the inferred latent and ground truth uses the geodesic distance on the ring (subsection D.9). **(B)** Applying regression (Poisson, Universal (X)) and joint latent-observed (Universal (X,Z)) models to the modulated Poisson population data. Left: three columns showing progressively how single neurons variability, with $\xi$ and $T_{\text{DS}}$ (middle), and noise correlations, with $r_{ij}$ and Fisher $Z$ (bottom), are captured (see subsection 2.4). Right: $\Delta$cvLL for all models (top) and visualization of the joint latent-observed model input spaces (middle) and tuning curves (bottom).

factors above and below 1). The second population consists of Poisson neurons tuned to head direction and an additional hidden signal, which gives rise to apparent overdispersion [28] as well as noise correlations when only regressing to observed covariates. For mathematical details of the count distributions and synthetic populations, see Appendix A and Appendix D, respectively.

We compare our UCM to the Poisson GP model [33] and the heteroscedastic negative binomial GP model, a non-parametric extension of [55]. To show data-efficiency and regularization benefits of

GPs, we also compare to a UCM with an artificial neural network (ANN) mapping replacing the GP mapping. For details of the baseline models, see Appendix D. For cross-validation we split the data into 10 roughly equal non-overlapping segments, and validated on 3 chosen segments that were evenly spread out across the data. When a latent space was present, we used $90\%$ of the neurons to infer the latent signal while validating on the remaining neurons, and repeated this for non-overlapping subsets. We rescale the log-likelihoods by the ratio of total neurons to neurons in subset and then take the average over all subsets to obtain comparable cross-validation runs to regression.

Figure 2A shows that only the UCM successfully captures the heteroscedastic CMP data, indicated by $T_{\mathrm{KS}}$. Baseline models cannot capture frequent cases where the Fano factor drops below 1. In addition, we observe that using a Bayesian GP over an ANN mapping in the model leads to a reduction in overfitting, especially in the latent setting (Figure 2A). Therefore, all other analyses with the UCM reported here used the GP mapping. Figure 2B shows that the modulated Poisson population activity is seen by a Poisson regression model as overdispersed, indicated by $T_{\mathrm{DS}}$. Our model flexibly captures the overdispersed single neuron statistics, independent from noise correlations $r_{ij}$ that are captured when we introduce a Euclidean latent dimension. As expected, the $\xi$ scatter plots shows whitening under the posterior predictive distribution when the correlations are captured.

## 3.2 Mouse head direction cells

We apply the UCM to a recording of 33 head direction cells in the anterodorsal thalamic nucleus (ADn) and the postsubiculum (PoS) of freely moving mice [41, 42], see subsection D.2. Neural data was binned into 40 ms intervals, giving $K = 11$. Note that observed count statistics differ with bin size, see subsection C.2 for a discussion. Regression was performed against head direction (HD), angular head velocity (AHV), animal speed, two-dimensional position in arena, and absolute time, which collectively form $X$ in this model. We used 64 inducing points for regression, and added 8 for every latent dimension added (Appendix E). Cross-validation was performed as in synthetic experiments, but with 6 validation segments and subsets of $85\%$ of neurons to infer the latent signal.

Figure 3A shows that for regression hNB overfits and performs worst, despite containing Poisson as a special case. However, this limit is not reached in practice due to the numerical implementation, see Appendix A. Only the UCM captures the training data satisfactorily with respect to confidence bounds for $T_{\mathrm{KS}}$ and $T_{\mathrm{DS}}$, although the data remained slightly underdispersed to the model with $T_{\mathrm{DS}}$ values slightly skewed to negative. Compared to the Poisson model, the cvLL is only slightly higher for the UCM as the data deviates from Poisson statistics in subtle ways. We see both FF above and below 1 (over- and underdispersed) across the neural firing range in Figure 3B, with quite some neurons crossing 1. Correspondingly, FF-mean correlations coefficients are often negative. Their spread away from $\pm 1$ indicates firing rate and FF do not generally satisfy a simple relationship, especially for examples such as cell 27. Furthermore, ADn neurons seem to deviate less from Poisson statistics. From Figure 3C, we note in particular that FFs tend to decrease at the preferred head direction, but rise transiently as the head direction approaches the preferred value. We also see that tuning to speed and time primarily modulates variability rather than firing rates. All of this is impossible to pick up with baseline models, which constrain FF $\geq 1$ as well as FF increasing with firing rate (Appendix A). Finally, we see more tuning of the firing rate to position in PoS cells.

When adding latent dimensions, Figure 3D shows a peak in the cvLL at two dimensions, where correspondingly the Fisher $Z$ distribution starts to match the unit normal well. Kernel length scales however did not indicate redundant latent subspaces for higher dimensions as expected for automatic relevance determination, likely due to mixing of latent dimensions. Notice the noise correlation patterns in Figure 3E tend to show positive correlations for similarly tuned neurons roughly around the diagonal of blocks, as expected from ring attractor models [22]. Intrinsic neuron variability, roughly quantified by the average FF, further decreased and thus become even more underdispersed when considering additional tuning to latents, in particular for ADn. In addition, latent signals primarily modulate firing rate as seen from TIs in Figure 3F. When looking at time scales of covariates in Figure 3G (computed as the decay time constant of the autocorrelogram (Appendix D), the latent processes seem to vary on time scales right in the gap of behavioural time scales.

Tangential to our main contribution of characterising the detailed structure of neural variability, our results have another novel element that does not specifically rely on the UCM. Using GP-based non-parametric methods, we successfully estimated the tuning of cells to as many as 8 different covariates (6 observed + 2 latent, see Figure 3G) in a statistically sound fashion, while previous

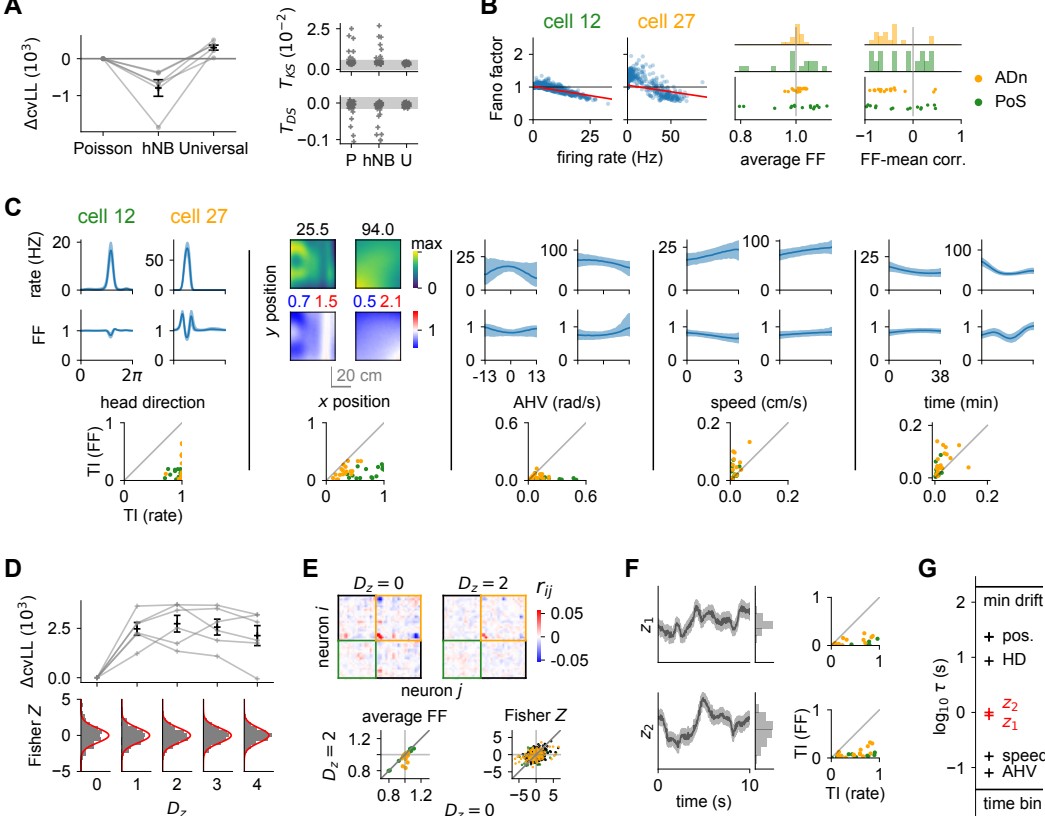

Figure 3: **Application to mouse head direction cells in the anterodorsal thalamic nucleus (ADn) and the postsubiculum (PoS). (A)** Goodness-of-fit for Poisson (P), heteroscedastic negative binomial (hNB) and universal (U) regression with covariates described in the text. Our method (U) outperforms baselines ($p = 7 \cdot 10^{-3}$ for $\Delta$cvLL w.r.t. Poisson, one-sample $t$-test). **(B)** Left: Fano factor (FF) versus mean count of the predictive distribution for all time steps (scatter dots) for two representative cells. Red lines are linear regression fits to scatter points. Right: average FF across time steps and Pearson $r$ correlation between FF and mean count for all cells, with colour indicating region ADn (orange) or PoS (green). **(C)** Top: conditional tuning curves (subsection D.5) of the UCM with complementary $x$ at preferred HD, positioned at centre of arena, with zero speed, AHV and at time $t = 0$. Bottom: TIs w.r.t. corresponding covariates for the FF and firing rates. **(D)** Adding Euclidean latent dimensions $D_z$. Top: $\Delta$cvLL w.r.t. regression only. Bottom: Fisher $Z$ histogram, with red curves its sampling distribution. **(E)** Comparison between models with latent space dimensions $Dz = 2$ and $D_z = 0$. Top: off-diagonal noise correlations $r_{ij}$, with neurons ordered by region first (PoS and ADn), and then by preferred head direction within region. Bottom: average FF (same as in (B)) and Fisher $Z$ under the posterior predictive distribution Equation 12. **(F)** Left: inferred latent trajectories for $D_z = 2$. Right: corresponding TIs for FF and firing rates. **(G)** Time scales for covariates computed from autocorrelograms (Appendix D). Note pos. refers to both $x$ and $y$ position (near-identical values). The top horizontal line is the minimum kernel length scale over absolute time across neurons. The bottom line depicts the 40 ms time bin. Error bars in **A** and **D** show s.e.m. over cross-validation runs. Shaded regions for $T_{KS}$, $T_{DS}$, tuning curves and latent states show the $95\%$ CI.

GP-based approaches typically only consider around 2 to 3 input dimensions [56, 33, 43]. Specifically, one of our covariates was absolute experimental time to capture non-stationarities in neural tuning. As a result, our model captured several experimental phenomena that are studied separately in the literature: drifting neural representations [57, 58, 59], anticipatory time intervals [54] and conjunctive tuning to behaviour [60]. We also applied the model in a purely latent setting similar to the example in Figure 2A, with the UCM uncovering a latent signal more closely correlated to the head direction compared to baseline models. These additional results are presented in Appendix C.

# 4 Discussion

**Related work**   Neural encoding model provide a statistical description of neural count activity, and typically rely on a parametric count likelihood, such as Poisson [33] or negative binomial [30, 31, 61], that is often mismatched to empirical count statistics. Heteroscedastic versions additionally regress the dispersion parameter of count distributions to covariates [62, 63]. This has shown improvements in stimulus decoding and more calibrated posterior uncertainties [55]. Copula-based models [36, 64] separate marginal distributions of single neurons from the multivariate dependency structure parameterized by the copula family, and thus remove parametric constraints on single neuron count statistics. Generally, the idea of a universal model that can capture arbitrary joint distributions has been explored for binary spike trains [37]. However, neither approach naturally incorporates modulation of spike count distributions by input covariates.

Our model deals with discrete spike counts ranging from $0$ to $K$ in a manner similar to categorical output variables in classification, where existing GP-based approaches pass function points directly through a softmax nonlinearity [65, 66]. Our approach instead relies on a basis expansion and linear-softmax mapping. At small time bins where $K = 1$, our model becomes identical to Bernoulli models [67] and comparable to point process models [51, 68, 69, 70]. In these cases, modulation of count variability becomes inseparable from firing rate modulation, making it difficult to generalize for heteroscedasticity in an interpretable manner and thus inconveniet for studying response variability. Introducing unobserved input variables incorporates aspects of Gaussian process latent variable models [71, 72]. Such models have been applied to neural data to perform dimensionality reduction [33], with extensions to non-Euclidean latent spaces and non-reversible temporal priors [43, 73].

**Limitations and further work**   The empirical choice of hyperparameters $C$ and basis functions $\phi$ is based on achieving sufficient model flexibility, as confirmed with the Kolmogorov-Smirnov approach. Recently, a multivariate extension of the Kolmogorov-Smirnov test has been proposed to directly test multivariate samples against the model [74], instead of looking at single neuron statistics. Alternatively, one could perform ARD [75, 76, 61] by placing a Gaussian prior on $W$, allowing automatic selection of relevant dimensions once a basis expansion is chosen. Another avenue for future work could consider going completely non-parametric by adding a count dimension to the input space, which is evaluated at counts $0$ to $K$ for every time step. This however increases the number of evaluation points by a factor $K + 1$. For high-dimensional input stimuli common in sensory areas, deep kernels [77] provide a scalable modification of our framework. In addition, extending our model with more powerful priors for latent covariates, such as Gaussian process priors [33, 73], can improve latent variable analysis, especially at smaller time bins where the temporal prior influence becomes more important. Regularization methods may help to decorrelate inferred trajectories [78, 79].

**Conclusion and impact**   We introduced a universal probabilistic encoding model for neural spike count data. Our model flexibly captures both single neuron count statistics and their modulation by covariates. By adding latent variables, one can additionally capture neural correlations with potentially interpretable unobserved signals underlying the neural activity. We applied our model to mouse head direction cells and found count statistics that cannot be captured with current methods. Neural activity tends to be less variable at higher firing rates, with many cells showing both over- and underdispersion. Fano factors and mean counts generally do not show a simple relation and can even be decoupled, with Fano factor modulation comparable or in some cases even exceeding that of the rate. Finally, we found that a two-dimensional latent trajectory with a timescale of around a second explained away noise correlations in these cells.

Neural variability is usually not considered on the same footing as mean firing rates, with models assigning most computational relevance to rates [80, 81]. However, recent work on V1 has started to explore variability as playing a computationally well-defined useful role in the representation of uncertainty [24, 25, 22, 26]. The framework introduced in this paper provides a principled tool for empirically characterising neural variability and its modulations – without the biases inherent in traditional approaches, which would likely miss potentially meaningful patterns in neural activities beyond mean rates. Our model has the potential to reveal new aspects of neural coding, and may find practical applications in designing and improving algorithms for brain-machine interfaces. As progress is made in scaling and applying such technology beyond research environments [82], it becomes increasingly more important to maintain transparency, e.g. through open source code, and to raise awareness of potential ethical issues [83].

## Acknowledgments and Disclosure of Funding

This work was supported by the Cambridge European and Wolfson College Scholarship by the Cambridge Trust (D.L.) and by the Wellcome Trust (Investigator Award in Science 212262/Z/18/Z to M.L). We are grateful to K.T. Jensen and A. Melkonyan for helpful feedback on the manuscript.

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
