# A universal probabilistic spike count model reveals ongoing modulation of neural variability
## Supplementary Material

## A    Parametric count distributions

### A.1    Poisson distribution

The Poisson count distribution is defined with a mean count $\lambda$

$$P_{\text{Poiss}}(n|\lambda) = \frac{\lambda^n}{n!}\, e^{-\lambda}. \tag{13}$$

where $n \in \mathbb{N}_0$. It describes a process where discrete events arriving in a time window are all independent of each other. Mathematically, this is consistent with Equation 13 being the limit of a binomial distribution $P_{\text{Bin}}(n, p)$ with $n \to \infty$ and $p \to 0$, such that $Np = \lambda$.

The Poisson distribution is characterized by the equality of its mean and variance, leading to a Fano factor $\mathrm{V}[n]/\mathbb{E}[n] = 1$. Another convenient property is that the sum of two independent Poisson processes is itself a Poisson process with $\lambda = \lambda_1 + \lambda_2$. This can be shown directly by considering $P(n) = \sum_0^n P(k)\, P(n-k)$ or casting it as a limit of the sum of two Bernoulli processes, and follows intuitively from the fact that spike times are independent of each other. As a consequence, for the inhomogeneous case where we have a time-dependent rate $\lambda(t)$ the count distribution over a longer interval is still Poisson with average $\int_0^T \lambda(t)\, \mathrm{d}t$. Note that this property does generally not hold for non-Poisson distributions, where the count distribution of a sum of counts in separate time windows is not related to the original count distribution in a simple way.

### A.2    Non-Poisson count distributions

To account for over- and underdispersed neural activity in real data, i.e. Fano factors above and below 1, other distributions than the Poisson count distribution have been used, and we present common families below.

#### A.2.1    Zero-inflated Poisson

A common way to introduce overdispersion is to model excess zero counts, which in this context leads to the zero-inflated Poisson (ZIP) process [1]. The count distribution is given by

$$P_{\text{ZIP}}(n|\lambda, \alpha) = \begin{cases} \alpha + (1 - \alpha)\, e^{-\lambda} & \text{if } n = 0 \\ (1 - \alpha)\, \frac{\lambda^n}{n!}\, e^{-\lambda} & \text{if } n > 0. \end{cases} \tag{14}$$

The parameterization leads to $\mathbb{E}[n] = \lambda(1 - \alpha)$ and $\mathrm{V}[n] = \lambda(1 - \alpha) + \lambda^2 \alpha(1 - \alpha)$ using the law of total variance.

#### A.2.2    Modulated Poisson distributions

One perspective of non-Poisson distributions is that they arise from noise in the rate parameters $\lambda$. Such count processes are referred to as modulated Poisson processes. From a probabilistic point of view, the resulting count distribution is a marginalization

$$P(n|\theta) = \int P(n|\lambda, \theta)\, p(\lambda|\theta)\, \mathrm{d}\lambda, \tag{15}$$

with noise parameters $\theta$. A recently proposed flexible spike count model that can give rise to different mean-variance relationships, including decreasing Fano factors at high firing rates similar to what is observed in Figure 4B, builds on this framework [2]. However, the modulated Poisson process can only account for overdispersion with respect to the base Poisson process. Adding noise cannot lead less variability here, and this implies that Fano factors are bounded from below by 1.

### A.2.3 Negative binomial

The negative binomial distribution is based on independent Bernoulli trials like the binomial distribution. However, now we count the number of successes before $r$ failures are observed. If we have Bernoulli trials with success probability $p$, one can obtain the negative binomial distribution with parameterization $p = \frac{\lambda}{r+\lambda}$

$$P_{\text{NB}}(n|\lambda, r) = \frac{\lambda^n}{n!} \frac{\Gamma(r+n)}{\Gamma(r)(r+\lambda)^n} \left(1 + \frac{\lambda}{r}\right)^{-r}. \tag{16}$$

Note this distribution is a specific instance of a modulated Poisson process (Equation 15), with $\lambda \sim f_{\text{Gamma}}(\lambda; r, \frac{\lambda}{r})$. The parameterization is such that $\mathbb{E}[n] = \lambda$ holds, but $\text{V}[n] = \lambda(1 + \frac{\lambda}{r})$ making it overdispersed with respect to a Poisson distribution. In practice, numerical evaluation of the Poisson limit when $r = 0$ is only approximate due to the numerical precision of the relevant function implementations.

### A.2.4 Conway-Maxwell-Poisson

A distribution that handles both over- and underdispersed count distributions is the Conway-Maxwell-Poisson distribution [3]

$$P_{\text{CMP}}(n|\lambda, \nu) = \frac{1}{Z(\lambda, \nu)} \frac{\lambda^n}{(n!)^\nu}. \tag{17}$$

The normalization constant has no closed form expression and must be evaluated numerically

$$Z(\lambda, \nu) = \sum_{k=0}^{\infty} \frac{\lambda^k}{(k!)^\nu}. \tag{18}$$

It contains the Bernoulli ($\nu \to \infty$), Poisson ($\nu = 1$) and geometric ($\nu \to 0$) distributions as limiting cases. The notably property is that the CMP distribution provides a smooth transition between these well-known distributions. At integer $\nu$, the The moments of this distribution do not have a closed form expression in general, but can be computed using the partition function through the cumulant generating function $K(t) = \log \mathbb{E}[e^{tn}] = \log Z(\lambda e^t, \nu) - \log Z(\lambda, \nu)$. The expression for the mean and variance follow to be

$$\mathbb{E}[n] = \lambda \frac{\text{d}}{\text{d}\lambda} \log Z(\lambda, \nu)$$
$$\text{Var}[n] = \lambda \frac{\text{d}}{\text{d}\lambda} \mathbb{E}[n]. \tag{19}$$

with approximate expressions [3]

$$\mathbb{E}[n] = \lambda^{1/\nu} + \frac{1}{2\nu} - \frac{1}{2}$$
$$\text{Var}[n] = \frac{1}{\nu} \lambda^{1/\nu}. \tag{20}$$

which hold well for $\nu \approx 1$ and $\lambda > 10^\nu$.

### A.3 Linear-softmax count distributions

The count distributions used in this work rely on a linear mapping of the input $\boldsymbol{a}$ combined with a softmax

$$P(n|\boldsymbol{a}; W, \boldsymbol{b}) = \text{softmax}(W\boldsymbol{a} + \boldsymbol{b}), \quad \text{softmax}(\boldsymbol{x})_i = \frac{e^{x_i}}{e^{\sum_j x_j}} \tag{21}$$

To illustrate the connection of this softmax count distribution used in Equation 1 to Poisson models, consider the distribution specified by the softmax mapping for $C = 1$ and the element-wise linear-exponential described in subsection 2.1, which in this case simply is $\phi(\boldsymbol{f}) = (f_1, e^{f_1})$. This choice contains the truncated Poisson distribution with $f$ as the logarithm of the mean count, corresponding to $W_{j0} = j$, $W_{j1} = -1$ and $b_j = 0$ with $\boldsymbol{a} = \phi(f)$. Hence for $C > 1$, our model is a generalization of rate-based models that implicitly assume neurons can be described by a single scalar rate parameter. The variability in such models is determined by a simple parametric relationship to the rate set by the count distribution, as can be seen for the count distribution families above.

# B  Neural dispersion and goodness-of-fit quantification

## B.1  Kolmogorov-Smirnov framework

The measures $T_{\text{KS}}$ and $T_{\text{DS}}$ introduced in subsection 2.4 provide a statistical goodness-of-fit measures of the model to single neuron count statistics, and is evaluated per neuron. The predictive count distribution of the model is the reference distribution for evaluating $\xi$ (Equation 7), and thus allows one to quantify dispersion $T_{\text{DS}}$ (Equation 11) and goodness-of-fit $T_{\text{KS}}$ (Equation 10) of the data with respect to our predictive model. By using a full predictive model, the Kolmogorov-Smirnov framework is applicable to data beyond repeatable trial structure in the inputs, such as continual recordings of freely moving animals.

For $T$ values $u_t$, $T_{\text{KS}}$ is defined as

$$T_{\text{KS}} = \max_t |F_T(u_t) - F(u_t)| \tag{22}$$

with cumulative distribution function $F(u)$ and empirical distribution function

$$F_T(u) = \frac{1}{T} \sum_{t=1}^{T} 1_{u_t \leq u} \tag{23}$$

If the $u_t$ are uniformly distributed $u \sim \mathcal{U}(0,1)$ in the null hypothesis or generative model, we have $F(u) = u$. This leads to the expression given in Equation 10, which is the Kolmogorov-Smirnov statistic relevant to this work. $T_{\text{KS}}$ has an asymptotic sampling distribution based on the Brownian bridge [4]. The unit Brownian bridge is defined for a Wiener process $W(t)$ as

$$B(t) = W(t) - W(1), \quad \text{for } 0 \leq t \leq 1 \tag{24}$$

and the sampling distribution corresponds to the distribution of $N^{-1/2} \sup_t B(t)$.

This statistic can be interpreted as an out-of-distribution score for the observed sample, with significant misfit when $T_{\text{KS}}$ is above significance value. Conventional statistics uses hypothesis testing to assess the model fit, with the null hypothesis being that the data is statistically indistinguishable from the predictive model. We can obtain model acceptance regions based on some cutoff significance value of the test statistic under its sampling distribution, often taken to be $5\%$. An alternative is to assess how close the empirical distribution of the test statistic is to the sampling distribution, which is the expected distribution of the statistic under the predictive model. This can be done with another Kolmogorov-Smirnov test. In this paper, we plot the acceptance regions of $T_{\text{KS}}$ and show them compared to baseline models to highlight the model fit improvement on the data it was fit on. $T_{\text{DS}}$ was treated similarly as a test statistic for measuring dispersion of the data with respect to the model. We present the asymptotic sampling distribution of $T_{\text{DS}}$ below in subsection B.3.

## B.2  Traditional variability measures

The traditional $Z$-score [5, 6, 7] and Fano factor [8, 2] have been used widely in the literature to quantify the variability in neural responses. The two measures are directly related

$$\text{FF} = \langle Z^2 \rangle \quad \text{with} \quad Z = \frac{y - \langle y \rangle}{\sqrt{\langle y \rangle}}, \tag{25}$$

with $y$ denoting spike counts and $\langle \cdot \rangle$ the average over the relevant set of trials or time segments of experimental data. Under Poisson spiking statistics, the Fano factor is 1 and the $Z$-score is distributed as a unit normal variable. Neural data with more or less variability will lead to deviations from this reference for these dispersion measures. Activity more variable than Poisson is called overdispersed, and vice versa for underdispersed activity. These measures are mostly applied to trial-based data [9], but they can also be applied across separate time windows within a given trial or run in continual recordings. In continuous tasks as free animal navigation, the $Z$-score is often used to quantify variability or dispersion [5, 7, 1].

Note the normality of $Z$-scores under Poisson data is only asymptotically true, in the sense that we require the predicted average count $\langle y \rangle \ll 1$. The generalized $Z$-score $\xi$ in Equation 7 are Gaussian under the true model by design, independent of the spike count distribution and count

magnitudes. However, segments with low expected spike counts around 1 are affected significantly by the dequantization noise, hence the normality in those cases is due to the dequantization rather than model fit.

From Equation 25, we can see that our definition of a dispersion measure $T_{\mathrm{DS}}$ in Equation 11 is mathematically almost identical to the log Fano factor with $Z$-scores replaced by $\xi$ (Equation 25). By using these generalized $Z$-scores, we can evaluate dispersion with respect to an arbitrary reference count distribution. However, the role of the two quantities are different. Fano factors are used to provide a measure of the spike count variability, with value 1 placing a reference point at Poisson statistics. On the other hand, $T_{\mathrm{DS}}$ is used to quantify whether the observed spike count dispersion is statistically significant compared to variability predicted by the model.

### B.3  The sampling distribution of $T_{\mathrm{DS}}$

Under the true model, generalized $Z$-scores $\xi$ (Equation 7) are i.i.d. Gaussian variables across neurons and time, hence the dispersion measure $T_{\mathrm{DS}}$ based on the sample variance of $\xi$ follows a $\chi^2$-distribution. More precisely, for i.i.d. Gaussian $\xi_i \sim \mathcal{N}(0,1)$, the population variance

$$s^2 = \frac{1}{N} \sum_i \xi_i^2 \tag{26}$$

has $Ns^2$ distributed as a $\chi^2$-distribution with $N$ degrees of freedom.

The moment generating function defined as $M(t) = \langle e^{-tX} \rangle_X$ is a useful quantity for computing the moments of a distribution $p(X)$. Note that $M^{(n)}(0)$, indicating the $n$-th derivative with respect to time, gives us $(-1)^n \langle X^n \rangle_X$. When we consider the asymptotic convergence to a normal distribution of the $\chi^2$-distribution, the distribution of $\log s^2$ has more favourable convergence property as it is less skewed due to the logarithmic transformation [10]. Its moment generating function is

$$
\begin{aligned}
M(t) &= \int_0^\infty (s^2)^{-t} \left( \frac{Ns^2}{2\sigma^2} \right)^{\frac{N}{2}-1} e^{-\frac{Ns^2}{2\sigma^2}} \frac{Ns}{\sigma^2} \, \mathrm{d}s \, / \, \Gamma\left( \frac{N}{2} \right) \\
&= \left( \frac{2\sigma^2}{N} \right)^{-t} \Gamma\left( \frac{N}{2} - t \right) / \Gamma\left( \frac{N}{2} \right)
\end{aligned}
\tag{27}
$$

which gives rise to the cumulant function

$$K(t) = \log M(t) = t \log \frac{N}{2} + \log \Gamma\left( \frac{N}{2} - t \right) - \log \Gamma\left( \frac{N}{2} \right) \tag{28}$$

From here we can compute the first two cumulants as $\kappa_n = K^{(n)}(0)$ similar to the moment generating function, which are equivalent to the mean and variance of the distribution

$$
\begin{aligned}
\mu &= \kappa_1 = \psi\left( \frac{N}{2} \right) - \log \frac{1}{2}N \\
\sigma^2 &= \kappa_2 = \psi'\left( \frac{N}{2} \right)
\end{aligned}
\tag{29}
$$

with $\psi(x) = \Gamma'(x)$ i.e. the first derivative of the Gamma function, and the notation $f'(x) = \mathrm{d}f(x)/\mathrm{d}x$. For values $N \gtrsim 20$, the following asymptotic expression hold well [10]

$$
\begin{aligned}
\mu &= -\left( \frac{1}{N} + \frac{1}{3N^2} \right) \\
\sigma^2 &= \frac{2}{N-1}
\end{aligned}
\tag{30}
$$

These properties lead to the construction of the $T_{\mathrm{DS}}$ metric in Equation 11. It has an asymptotically normal sampling distribution with mean 0 and variance $2/N-1$, convenient for statistical testing and confidence intervals.

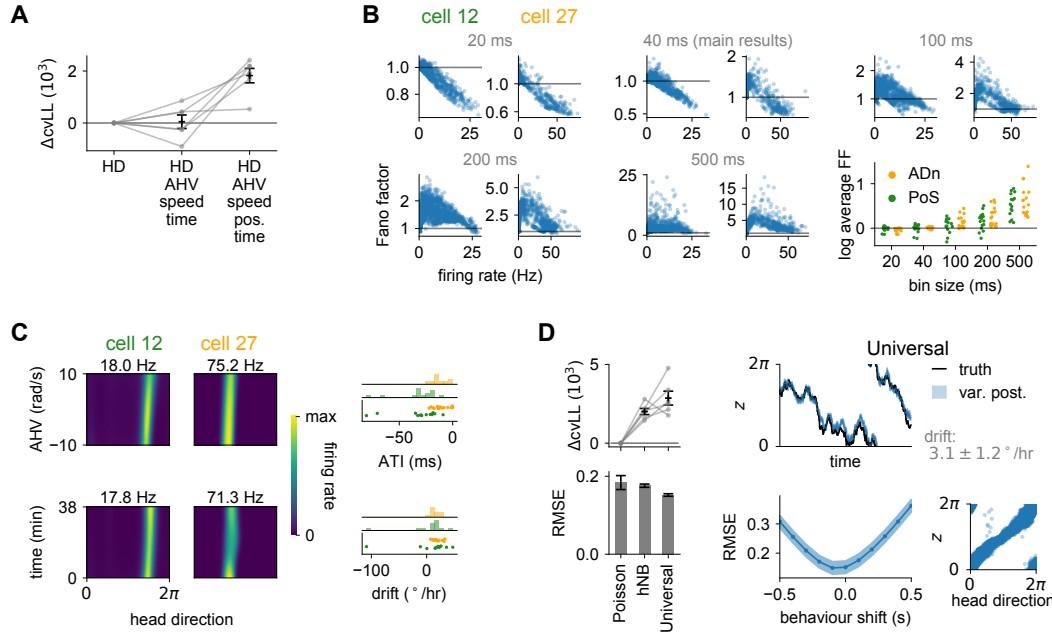

Figure 4: **Additional analysis on head direction data. (A)** Model comparison of the UCM with different regressors (HD: head direction, AHV: angular head velocity, speed: running speed, pos.: animal position, time: experimental time). $\Delta$cvLL shows cross-validated log-likelihood w.r.t. the model with only HD as the regressor. **(B)** Fano factor and mean counts as predicted by the mean posterior count distribution with each dot representing one time step, similar to Figure 3B. We show examples for two representative cells, from model fits at various bin sizes (20, 40, 100, 200 and 500 ms). On the bottom right, we plot the log of the mean Fano factor across all time steps against the bin size for all cells in the data recorded in ADn (orange) and PoS (green). **(C)** Left: Joint conditional tuning curves of firing rate in the full regression model (with complementary covariates fixed at the same values as in Figure 3C) as a function of AHV-HD (top) and time-HD (bottom) show two distinct experimental phenomena: anticipatory tuning and neural representational drift, respectively, for two representative cells. Right: anticipatory time intervals (ATI, top) and drifts (bottom) as defined in subsection C.3, with corresponding histograms, for selected cells (subsection D.8) in the data from ADn (orange) and PoS (green). **(D)** Application of the UCM with only latent regressors. Left: $\Delta$cvLL with respect to Poisson baseline (top) and root-mean-squared error (RMSE, bottom) of estimated latent variable w.r.t. true head direction for different variants, using a Poisson, heteroscedastic negative binomial (hNB), and universal likelihood. Our model (Universal) performs best ($p = 1.3 \cdot 10^{-3}$ for $\Delta$cvLL, one-sample $t$-test). Right: experimentally observed head direction (black) and estimated latent variable (blue; shaded region shows the $95\%$ CI) in the fitted UCM model as a function of time (top left) and compared against each other (bottom right), matched by fitting a constant angular offset, a sign reversal, and linear drift in time (top right, grey; see Equation 45). We also temporally shifted observed head direction (behaviour) w.r.t. the latent signal, and repeated the same fitting procedure to compute the cross-validated RMSE (bottom left, shaded region shows s.e.m. over cross-validation runs). Error bars in **A** and **D** show s.e.m. over cross-validation runs.

## C  Additional analysis of head direction cells

### C.1  High-dimensional behavioural input

The universal count model (UCM) was regressed against 6 input dimensions in Figure 3A. Such high-dimensional input spaces are rife with undersampled regions, and to check if the model overfit to data we fit UCMs with a smaller number of input dimensions as shown in Figure 4A. Adding more regressors starting from only head direction progressively improves $\Delta$cvLL, indicating the model did not overfit in the full regression case considered in this paper.

## C.2 Temporal bin sizes

In the limit of very small time bins and $K = 1$, our model becomes a generalisation of the universal binary model [11], allowing for (observed and latent) covariates – which is critical for dissecting signal and noise correlations in neural data. In fact at $K = 1$, the distinction between classical models assuming Poisson variability and our universal model allowing for non-Poisson variability becomes irrelevant as all possible spike count distributions (SCDs) in a time bin are Bernoulli. Thus, in this limit, our model also becomes conceptually very similar to GLMs [12] and GPFA [13] (with binary emissions) in that conditioned on covariates (potentially including the spiking history of a neuron itself, or of other neurons, in GLMs) spiking becomes an inhomogeneous Poisson process. However, these previous models remain Poisson-like *at all time scales* (in the sense that, conditioned on covariates, spike counts remain Poisson distributed) and only allow covariates to modulate the mean firing rate. This is because, given covariates, spiking is assumed to be independent in consecutive infinitesimal time bins, a key property of the Poisson point process. The key advance of our work is precisely that *even at larger time bins* (and with $K > 1$) it is not restricted to Poisson count distributions and allows covariates to modulate any spike count statistic. Indeed, we found that experimental data deviated from Poisson-like statistics in important ways and was in many cases substantially modulated by covariates at the $40$ ms time bin size (with maximum spike count $K = 11$) that we chose (Figure 3B and C).

The advantage of choosing essentially infinitesimally small time bins is that there is no "arbitrary" (though see above) time bin size parameter and every individual spike can be predicted (at least in principle, see also note below). The disadvantage is that, as we explain above, in this case it is not even conceptually possible to model modulations of response variability independent from modulations of firing rates, as the two are inseparable in the underlying Bernoulli model (as $K = 1$). Indeed, experimental studies of neural variability, and its modulation by covariates, have always used larger time bins to compute Fano factors or Z-scores [14, 1, 5, 9]. Our work is a direct generalization of this perspective, extending beyond rigid trial-structure. At the other extreme, the disadvantage of choosing time bins that are too large is that it may miss the time scale at which covariates modulate neural firing. In particular, when studying phenomena at time scales comparable to the interspike intervals, such as theta phase precession [15, 16], binning may average away such effects if the bin size is too large. However, binning does reduce the number of total time points and is thus more practical for studying the activity of large populations recorded over long time periods.

Our choice of $40$ ms time bins was based on previous empirical measures of autocorrelation time scales in neural activity [17, 18]. We also ensured none of our behavioural covariates had a shorter time scale (see Figure 3G). In Figure 4B, we can see the sensitivity of the count analysis to bin size. We fit separate UCMs to data at various bin sizes as presented in the figure. Note that increasing the bin size leads to higher Fano factors. In addition, notice the consistent decrease of variability at higher firing rates. Temporal correlations in the spike trains generally lead to spike counts being correlated across consecutive time bins. This in turn leads to potentially more extreme fluctuations in the sums of consecutive counts (qualitatively identical to picking a larger bin size), and thus higher or lower variability. The exact details depend on the underlying process, and in general no analytical treatment is possible. For stationary renewal processes however, an analytical treatment of Fano factor dependence on bin size is available [19].

In summary, there is nothing to say *a priori* that *the right* time bin size for studying neural variability is the infinitesimally small limit used by several previous approaches. Indeed, our empirical results showing non-Poisson conditioned SCDs at longer time scales suggest that longer time scales may be more appropriate – at least in the data set we analysed. In general, we argue (see above) that if one wants to study the modulation of neural response variability then one must use appropriately sized (non-infinitesimally small) time bins. In turn, in this setting, our approach is unique in offering a statistically principled method to do so and offers novel insights into the variability of head direction cells in mice.

## C.3 Drifting and ATIs

Joint tuning curves can reveal neural representations that are not factorized over a set of covariates. The Bayesian nature of Gaussian processes takes care of undersampled regions that are rife in high dimensional input spaces, which is the setting in this work for studying joint tuning to high-

dimensional behavioural input. By looking at joint tuning curves between particular covariates, the model reveals properties that have been observed in separate experimental works.

One can pick up representational drift [20, 21] by regressing against absolute time of the recording. We can then compute a linear drift by quantifying how much the preferred head direction $\theta_{\mathrm{pref}}$ (Equation 40) changes with time using circular linear regression (subsection D.8). Not all cells are well-described by a linear drift in time, and only cells that have a sufficiently good fit to the regression line are included. The time regressor needs to be considered carefully as it may confound at time scales of latent trajectories. As long as the time scale in the kernel is much larger than the time scale over which the latent variables vary, we can interpret the temporal drift as a separate process from the latent trajectories. Indeed, by initializing at time scales equal to half the total recording time, these time scales of the Gaussian process kernel remain significantly higher than any behavioural time scale (Figure 3G). We find most cells cluster at a drift of $\approx 20$ °/hr.

The joint AHV-HD plot reveals anticipatory tuning: when animals turn their head, the head direction tuning curves shift in response to head rotations such that cells expected to spike appear to fire earlier than expected. Theoretical studies have shown that this improves temporal decoding, in the sense that the bias-variance trade-off for decoding downstream can be improved with anticipatory tuning [22]. It appears that the head direction population anticipates the future head direction based on current movement statistics, which allows one to reduce the bias introduced with causal decoding. One can define an anticipatory time interval (ATI) analogous to linear drift, as the amount of change in preferred head direction with angular head velocity. This is again quantified using circular regression (subsection D.8). Similarly, not all cells are well-described by a linear relation between $\theta_{\mathrm{pref}}$ shift and angular head velocity $\omega$, and only cells that have a sufficiently good fit to the regression line are included. Note the ATI values in Figure 4C are negative, while in the literature they are postive and differ per region. The neural data description files [23] did mention that the zero time frame of behaviour was randomly misaligned to neural spiking data up to 60 ms. Behaviour may be shifted with respect to the neural spike train, indeed in preliminary analyses with shifted spike trains we found values consistent with literature for ATIs when shifting $\approx 60$ ms [22].

### C.4 Latent variable analysis of head direction data

In Figure 4D, we see the inferred angular latent signal is closely related to the head direction. This is similar to previous analyses with non-Euclidean Gaussian process latent variable models [24] and presents an exceptional case where the inferred latent signal is directly relatable to an experimentally observed variable. Therefore, a measure of error can be computed between the two. To do so, we align the latent signal to the observed head direction by fitting a transformation of the form described in Equation 45. Latent trajectory root-mean-squared error (RMSE) was computed with 3-fold cross-validation, using the geodesic on the ring. We align the latent trajectory Equation 45 to the behaviour in the fitting segment, and compute the geodesic RMSE on the held-out validation segment (see subsection D.8).

The UCM again shows improvement over baseline models, and interestingly the inferred latent signal is more correlated to the behavioural head direction (left bottom panel of Figure 4D). When shifting the behaviour w.r.t. the latent signal in the UCM model, we observe a minimum cross-validated RMSE at around $-100$ ms. We thus tentatively identify a delay in the signal represented compared to measured behaviour, with the behaviour lagging the latent signal. From the UCM fit at zero behavioural shift, the inferred linear drift compared to observed head direction is $3.1 \pm 1.2$ °/hr, see right top panel of Figure 4D. This is smaller but in the same direction as the drift found using regression in the tuning curves in panel C, which cluster around 20 °/hr.

## D  Analysis details

### D.1  Synthetic data

We construct a synthetic head direction cell population inspired by bump attractor models [25, 26, 24]. Firing rate tuning curves to head direction $\theta$ are parameterized as von Mises bump functions with some constant offset

$$f(\theta; b, A, \beta, \theta_0) = A\, e^{\beta\, \cos\,(\theta - \theta_0)} + b \tag{31}$$

with $b > 0$ and $A > 0$. This results in $f \geq 0$ for all valid inputs and parameters. For modelling firing rates, we additionally restrict ourselves to $\beta > 0$ to avoid inverted bumps at the preferred head direction $\theta_0$.

In the Conway-Maxwell-Poisson (CMP) synthetic population, we placed the tuning curves from Equation 31 on parameters $\nu$ and the approximate mean $\mu_y = \mathbb{E}[y]$ in Equation 20. Note both parameters have to be non-negative to be valid. Furthermore, the tuning curves of $\nu$ had potentially negative $\beta \in \mathbb{R}$ and different parameter statistics than for $\mu_y$. Again, these were chosen such that firing rates and variability were within the physiological regime. To roughly match the mean counts with von Mises bump pattern amplitudes, we used the mapping

$$\lambda = \left( \mu_y - \frac{1}{2\nu} + \frac{1}{2} \right)^{\nu} \tag{32}$$

which was based on the approximate relation Equation 20 of the mean. We chose a time bin of 100 ms, which led to $K = 18$ in the synthetic data generated. To sample from the CMP distribution once we specified $\lambda$ and $\nu$, we used the fast rejection sampling method [27].

For the modulation by a hidden Euclidean signal in the modulated Poisson population, we additionally placed Gaussian tuning curves on the latent dimensions with varying standard deviations and means. The Gaussian tuning curves tiled the latent space that was traversed, which allowed the model to infer the full trajectory. Note that tuning is factorized across the two dimensions (head direction $x$ and latent signal $z$). Parameters were randomly sampled from distributions that led to firing rates and variability within the physiological regime. We again picked a 100 ms time bin, which gave $K = 28$.

## D.2 Neural data

Data was taken from Mouse 28, session 140313, during the wake phase [23]. The spiking data was recorded at a resolution of 20000 Hz, whereas behaviour was extracted from video recordings of animal body tracking at a resolution of 39.06 Hz. Note the time of the first video frame was randomly misaligned by 0–60 ms to the neural spike trains. We removed invalid behavioural segments in the data and performed linear interpolation across those segments. For circular variables, interpolation was taken in the shortest geodesic distance. We binned spiking data at 1 ms, and interpolated behavioural data to reach the same sampling frequency that is higher than the behavioural recording frequency. At a binning of 40 ms used in our analysis, we had $K = 11$ as the maximum count value.

We selected head direction cells based on a sparsity criterion, after trying several criteria as mutual information typically used for place cells [28]. First, we binned the head direction variable into 60 equal bins over the range $[0, 2\pi]$. For each bin, we now compute the average spike counts $y_i$ for head directions within bin $i$, and the relative occupancy $P_i$. Note $\sum_i P_i = 1$ is a probability distribution. Sparsity is defined as

$$1 - \frac{(\sum_i P_i \, y_i)^2}{\sum_i P_i \, y_i^2} \tag{33}$$

and with a selection criterion of sparsity $\geq 0.2$ we obtained 33 head direction cells, of which 15 are in postsubiculum. Alternatively, although more computationally intensive, we could directly regress a Gaussian process model (e.g. Poisson baseline model Equation 34) and look at the kernel lengthscales on the angular input dimension. These will be appreciably larger than $2\pi$ for cells that are not tuned much to head direction.

Note that quite a few head direction units, which are supposed to represent single cells, show bimodal tuning curves or more to head direction. This is likely due to multiple neurons as signals can pollute in electrophysiological recordings and spike sorting can fail to distinguish between them [29, 30].

## D.3 Baseline models

### D.3.1 Gaussian process models

The log Cox Gaussian process model puts a GP prior on the rate function of an inhomogeneous Poisson process (Equation 13) with an inverse link function $f(x) = e^x$ that is exponential

$$
\begin{aligned}
h(x) &\sim \mathcal{GP}(\mu_x, k_{xx}) \\
\lambda(x) &= f(h(x)) \\
y &\sim P_{\text{Poiss}}(y | \lambda \cdot \Delta, \theta)
\end{aligned}
\tag{34}
$$

where the time bin length is $\Delta$, which turns $\lambda$ into a proper rate quantity.

The heteroscedastic negative binomial model builds on this encoding model, More precisely, two GPs with an exponential inverse link function are used to model tuning to covariates of the rate $\lambda$ and inverse shape $1/r$ of the negative binomial likelihood (Equation 16), leading to the model

$$h(x) \sim \mathcal{GP}(\mu_x, k_{xx}), \quad g(x) \sim \mathcal{GP}(\mu_x, k_{xx})$$

$$\lambda(x) = f(h(x)), \quad \frac{1}{r} = f(g(x)) \tag{35}$$

$$y \sim P_{\mathrm{NB}}(y|\lambda \cdot \Delta, r)$$

In the same spirit, we could construct the more flexible heteroscedastic Conway-Maxwell-Poisson model. This model would be able to capture both over- and underdispersed count data (Fano factors above and below 1), but it has difficulty in scaling to large data due to the series approximation of the partition function in Equation 17.

### D.3.2 Artificial neural network models

The artificial neural network (ANN) model used to replace the Gaussian process in validation experiments was designed such that there was sufficient expressivity to model the neural activity. In fact, we see that the neural network overfits in Figure 2A, which indicates that there was enough capacity in the network. The network architecture consists of an input layer providing $\boldsymbol{x}_t$ (and latent $\boldsymbol{z}_t$ when present), encoding angular dimensions $\theta$ as a two-dimensional vector $(\cos\theta, \sin\theta)$. There are 3 hidden layers containing 50, 50 and 100 hidden units in order from input to output layer, with sinusoidal activation functions to construct smooth overall mappings [31]. The output layer consisted of $N \cdot C$ linear units providing $f_{cnt}$ in Equation 1, with $N$ the number of neurons and $C$ the number of degrees of freedom per neuron (which was 3 in this work). The UCM with an ANN mapping leads to a model similar to VAEs [32] with a softmax likelihood and free variational parameters instead of amortization with an inference network.

### D.4 Computing generalized $Z$-scores

The generalized $Z$-scores $\xi$ in Equation 7 provide a normalized quantification of neural activity under the predictive model. For UCM, the count distribution $P(y)$ which is used to compute $\xi$ is taken to be the mean posterior count distribution of the posterior $q(\Pi|X, Z)$. In the case of baseline models, the reference $P(y)$ is given by the parametric distribution (Poisson in Equation 13, negative binomial in Equation 16) evaluated at the mean posterior values of the count distribution parameters given by the Gaussian process mapping (see Equation 34 and Equation 35). This is strictly speaking different from the mean posterior count distribution, as the parametric distribution depends nonlinearly on these parameters. However, the difference is insignificant when the variational uncertainties are small, which was the case in practice.

### D.5 Marginal and conditional tuning curves

Due to the high dimensional input space, we can either visualize slices of the tuning curve over the relevant input variables $\boldsymbol{x}_*$ or instead marginalize over other input variables. The conditional tuning curves are based on the count distributions $P(y|\boldsymbol{x}_*, \boldsymbol{x}^c)$, where $\boldsymbol{x}^c$ are fixed and cover the dimensions complementary to $\boldsymbol{x}_*$ (these are plotted in Figure 3C). On the other hand, marginalizing over $\boldsymbol{x}^c$ depends on the chosen $p(\boldsymbol{x}^c)$. A natural perspective is to consider the input data distribution $p_{\mathcal{D}}(\boldsymbol{x})$ and treat the observed input time series as a Markov Chain Monte Carlo path sampled from it. We can use this to approximate the exact marginalization, which is intractable as we do not know $p_{\mathcal{D}}(\boldsymbol{x})$. Conceptually, this is equivalent to an experimenter only looking at neural tuning to $\boldsymbol{x}_*$, which automatically marginalizes over all other behaviour not included during the experiment. We denote observed input with a subscript $X_{\mathcal{D}}$ in this scenario, to distinguish it from chosen input locations. We only consider the $\boldsymbol{x}^c$-dimensions of the joint density $p_{\mathcal{D}}(\boldsymbol{x})$ and this mathematically becomes

$$P(y|\boldsymbol{x}_*) = \int P(y|\boldsymbol{x}_*, \tilde{\boldsymbol{x}}^c) \, p_{\mathcal{D}}(\tilde{\boldsymbol{x}}) \, \mathrm{d}\tilde{\boldsymbol{x}} \approx \sum_t P(y|\boldsymbol{x}_*, (\boldsymbol{x}_{\mathcal{D}}^c)_t) \tag{36}$$

which defines the marginalization through the computation done in practice (summing over the time series of observed $\boldsymbol{x}^c$ while keeping $\boldsymbol{x}_*$ fixed). From this marginalized distribution, we can compute similarly quantities like the mean spike count or count variance.

For the tuning indices, we evaluate the the count statistic $T_y(\boldsymbol{x}_*)$ with respect to the posterior mean distribution $P(y|\boldsymbol{x}_*)$ after marginalizing (order does not matter as both are sums) to compute the tuning indices as described in Equation 6. Optimization over $\boldsymbol{x}_*$ of $T_y(\boldsymbol{x}_*)$ is done by grid search, as $\boldsymbol{x}_*$ is low-dimensional and we compute its values over a grid anyway for plotting tuning curves of mean, Fano factor or any other count statistic.

We used 300 Monte Carlo samples from $q(\Pi|X, Z)$ to compute the conditional tuning curves plotted in this paper. For marginalized tuning curves, we use 100 MC samples and temporally subsampled the observed input $X_{\mathcal{D}}$ to retain the first time step per every 10 time steps, and used this to evaluate Equation 36. As behaviour shows strong temporal correlations at short time scales (Figure 3G), this allows us to estimate the marginal tuning curves more efficiently. The mean of these samples was used to compute the mean posterior tuning curves for evaluating the TIs. When evaluating the average mean count and Fano factor at every time step (Figure 3B and Figure 4B), we used 10 MC samples from $q(\Pi|X, Z)$. When latent variables were present (Figure 3E), the 10 MC samples were drawn from $q(Z)$, corresponding to $m = 10$ and $k = 1$ in Algorithm 1.

### D.6  Temporal cross-correlations of covariates

We use the cross-correlation between time series $x_t$ and $y_t$

$$r_{xy}(\Delta) = \frac{\langle (x_{t+\Delta} - \langle x_{t+\Delta} \rangle)(y_t - \langle y_t \rangle) \rangle}{\sigma_x \sigma_y} \tag{37}$$

which includes the auto-correlation as a special case, e.g. $r_{xx}(\Delta)$. When one of the variables is a circular variable $\theta_t$, we use the linear-circular correlation coefficient in [33]

$$s_t = \sin \theta_t, \quad c_t = \cos \theta_t$$
$$R_{xs} = r_{xs}(\Delta), \quad R_{xc} = r_{xs}(\Delta), \quad R_{cs} = r_{cs}(\Delta)$$
$$r_{x\theta} = \frac{R_{xs}^2 + R_{xc}^2 - 2\,R_{xs}\,R_{xc}\,R_{cs}}{1 - R_{cs}^2} \tag{38}$$

and for the case when both are circular, we use the circular correlation coefficient proposed by [34]

$$s_\theta = \sin\left(\theta_t - \operatorname{Arg}\mathbb{E}[e^{i\theta_t}]\right), \quad \text{same for } \phi$$
$$r_{\theta\phi}(\Delta) = \frac{\mathbb{E}[s_\theta \cdot s_\phi]}{\mathbb{E}[s_\theta^2]^{\frac{1}{2}}\mathbb{E}[s_\phi^2]^{\frac{1}{2}}} \tag{39}$$

Time scales are estimated from the auto-correlations of covariates. The time scale $\tau$ is then chosen as the time step at which the value of the auto-correlation dropped by a factor $e$ from 1 at $\Delta = 0$.

### D.7  Preferred head direction

To compute the preferred head direction $\theta_{pref}$, we use the centre-of-mass of the firing rate profile $r(\theta)$ of head direction $\theta$

$$\theta_{pref} = \operatorname{Arg}[r(\theta)e^{i\theta}] \tag{40}$$

which is more robust to noise than taking the angle at which $r(\theta)$ is at a maximum. We can evaluate $\theta_{pref}$ as a function of angular head velocity (AHV) and absolute time to compute the ATIs and the neural drift as described in subsection C.3.

### D.8  Circular-linear regression

We computed the circular-linear regression [35] using a measure of the correlation between circular variables $\theta_1$ and $\theta_2$

$$R = |\mathbb{E}[e^{i(\theta_1 - \theta_2)}]| \tag{41}$$

By computing $R$ between a circular-linear function $\phi(t)$

$$\phi(t) = 2\pi a t + b \tag{42}$$

and the circular data time series $\theta_t$, we can perform the regression by maximizing $R$ through optimizing the parameter $a$ with gradient descent. The offset $b$ is obtained analytically

$$b = \operatorname{Arg}\mathbb{E}_t[e^{i(\theta_t - \phi(t))}] \tag{43}$$

From the values $a$ after fitting, one can compute the linear drift values and ATIs as described in subsection C.3. In addition, not all cells are well-described by the linear drift or ATIs, so we discarded cells which had an optimized value of $R < 0.999$. This cutoff was chosen as it retains cells that are visually in agreement with linear relations as seen in Figure 4, while discarding a few outlier cells.

### D.9 Latent alignments

To align 1D circular latent trajectories $z_c$ to a target trajectory, we minimize their mean geodesic distance under a constant shift $\mu$ and potential sign flip $s = \pm 1$

$$\tilde{z}_c = s \cdot z_c + \mu \tag{44}$$

We add a linear drift $\Delta$ to find potential drifting of the inferred trajectory

$$\tilde{z}_c = s \cdot z_c + \mu + t \cdot \Delta \tag{45}$$

as done in panel D of Figure 4. This is similar to the circular-linear regression above [35], but with the geodesic distance on the ring instead. This is consistent with root-mean-square errors in the latent signal from behaviour that are computed with the geodesic distances. For 1D Euclidean latent trajectories, we align by fitting a translation and scaling parameter.

In all cases, the root mean squared error (RMSE) of the alignment is evaluated in a cross-validated manner. For circular variables, we use the geodesic distance for computing the squared error just as in aligning. In more detail, we fit the trajectory transformation parameters such that we minimize the errors on the validation segment, and then use these fitted parameters to compute the transformed latent trajectory in the held-out segment. This is then used to compute the RMSE for the alignment of the cross-validation fold.

## E    Implementation details

### E.1    Mathematical details of the optimization objective

#### E.1.1    The sparse Gaussian process posterior

Exact Gaussian processes (GPs) have $O(T^3)$ computational complexity and $O(T^2)$ memory storage with $T$ input points [36]. This is unfavourable for scaling to large or massive datasets. In addition, non-Gaussian likelihoods lead to intractable marginal likelihoods and hence one needs an approximate optimization objective. Stochastic variational inference [37] provides a framework for applying Gaussian process methods using non-Gaussian likelihoods and approximations for scalability. Let us denote the exact GP prior by $p(\boldsymbol{f})$ with vector $\boldsymbol{f}$ the latent function points at the input locations $X$, the likelihood by $p(\boldsymbol{y}|\boldsymbol{f})$ with observed data $\boldsymbol{y}$, and the approximate posterior by $q(\boldsymbol{f})$. One then needs to be able to (1) efficiently and differentiably sample from $q(\boldsymbol{f})$, and (2) efficiently evaluate and differentiate the Kullback-Leibler (KL) divergence between $q(\boldsymbol{f})$ and $p(\boldsymbol{f})$.

Sparse approximations [38] reduce the computational complexity to $O(MT^2 + M^3)$ and storage to $O(M^2)$ with $M$ inducing points, which effectively aim to summarize the input data with a smaller set of points. Such methods are scalable for large $T$ as long as $M \ll T$ provides sufficient modelling flexibility. The key idea is to extend the function values $\boldsymbol{f}$ with additional function values $\boldsymbol{f}_u$ at inducing points. Let us denote inducing points with function values $\boldsymbol{f}_u$ at inducing point locations $U$, which we jointly learn with other variational parameters. The GP kernel evaluated at function point locations is denoted by $K_{XX}$, and at inducing point locations by $K_{UU}$. Cross-covariances are denoted by $K_{XU}$ and $K_{UX}$. The joint variational distribution to the augmented Gaussian process posterior $p(\boldsymbol{f}, \boldsymbol{f}_u|\boldsymbol{y})$ is defined as

$$q(\boldsymbol{f}, \boldsymbol{f}_u) = p(\boldsymbol{f}|\boldsymbol{f}_u)\, q(\boldsymbol{f}_u) \tag{46}$$

where the variational distribution $q(\boldsymbol{f}_u) = \mathcal{N}(\boldsymbol{m}, S)$, and $p(\boldsymbol{f}|\boldsymbol{f}_u)$ is the conditional Gaussian distribution from the generative model. The variational distribution over GP function values $q(\boldsymbol{f})$ is simply obtained by marginalizing out $\boldsymbol{f}_u$, which leads to a Gaussian with

$$\begin{aligned}
\mathbb{E}_q[\boldsymbol{f}] &= K_{XU}K_{UU}^{-1}\boldsymbol{m} \\
\text{Cov}_q[\boldsymbol{f}] &= K_{XX} - K_{XU}K_{UU}^{-1}K_{UX} + K_{XU}K_{UU}^{-1}SK_{UU}^{-1}K_{UX}
\end{aligned} \tag{47}$$

and the KL divergence

$$D_{\text{KL}}(q(\boldsymbol{f}, \boldsymbol{f}_u) || p(\boldsymbol{f}, \boldsymbol{f}_u)) = D_{\text{KL}}(q(\boldsymbol{f}_u) || p(\boldsymbol{f}_u)) = D_{\text{KL}}(\mathcal{N}(\boldsymbol{m}, S) || \mathcal{N}(\boldsymbol{0}, K_{UU})) \qquad (48)$$

which can be evaluated as long as $M$ is not too large. The reason for the choice in Equation 46 becomes clear: due to the cancellation of $p(\boldsymbol{f}|\boldsymbol{f}_u)$, we do not have to invert large matrices related to $K_{XX}$. Note that sampling from $q(\boldsymbol{f})$ for a large input set $X$ is problematic [39]. Fortunately, our likelihood factorizes across time and thus we can evaluate the expectation under $q(\boldsymbol{f})$ with the diagonalized distribution for which sampling is trivial.

Unlike purely variational approaches, the approximate posterior in Equation 46 amortizes the inference through the learned inducing points, and allows one to obtain a predictive distribution using the approximate posterior evaluated at a new set of inputs $X_*$

$$q(\boldsymbol{f}_*) = \int p(\boldsymbol{f}_* | \boldsymbol{f}_u) \, q(\boldsymbol{f}_u) \, \mathrm{d}\boldsymbol{f}_u \qquad (49)$$

This property also allows one to apply mini-batching or subsampling to Gaussian processes [40, 41]. Overall, this leads to the Sparse Variational Gaussian Process (SVGP), combinining Sparse Gaussian Processes [38] with stochastic variational inference [32].

To accelerate convergence, a different parameterization of the variational distribution is used. One performs a change of variables $\boldsymbol{v} = L_{UU}^{-1} \boldsymbol{f}_u$ with $L_{UU} L_{UU}^T = K_{UU}$ from the Cholesky decomposition. This transforms $p(\boldsymbol{f}_u)$ into $p(\boldsymbol{v}) = \mathcal{N}(\boldsymbol{0}, I)$, referred to as whitening, and the variational parameters are now defined for $q(\boldsymbol{v}) = \mathcal{N}(\boldsymbol{m}_v, S_v)$ [41]. In practice, matrix-vector products with $K_{UU}^{-1} = (L_{UU} L_{UU}^T)^{-1}$ are evaluated by solving two triangular systems with $L_{UU}$. The whitened representation simplifies Equation 47 as we do not need to compute $L_{UU}^{-1} \boldsymbol{m}$ and $L_{UU}^{-1} S (L_{UU}^{-1})^T$ in the non-whitened parameterization. The KL divergence Equation 48 also simplifies as we now have unit normal $p(\boldsymbol{v})$.

To increase the expressivity of multi-output GPs, a separate set of inducing points locations is used for each output dimension (neuron in this work), along with separate kernel hyperparameters as lengthscales for each input and output dimension. This is equivalent to modelling each output dimension by a separate GP, and leads to an overall computational complexity of $O(NCTM^2)$ and storage of $O(NCM^2)$ for our model (see section 2 for definition of quantities). A thorough description of a scalable multi-output SVGP framework is given in [42]. We define the multi-output variational posterior $q(F)$ as

$$q(F|X, Z) = \prod_{n=1}^{N} \int p(\boldsymbol{f}^{(n)} | \boldsymbol{f}_u^{(n)}) \, q(\boldsymbol{f}_u^{(n)}) \, \mathrm{d}\boldsymbol{f}_u^{(n)} \qquad (50)$$

with output function values $F$ evaluated at input locations $X, Z$.

### E.1.2 Generative model and variational inference

The overall generative model Equation 1 as depicted in Figure 1 is

$$P_\theta(Y|X) = \int \int P(Y|\Pi) \, p_\theta(\Pi|X, Z) \, p_\theta(Z) \, \mathrm{d}\Pi \, \mathrm{d}Z \qquad (51)$$

with the product of individual count distributions $P(Y|\Pi)$. The model parameters $\theta$ include the GP $\theta^{\text{GP}}$ and the prior $\theta^{\text{pr}}$ (hyper)parameters, as well as the softmax mapping weights $W_n$ and biases $\boldsymbol{b}_n$. Note that the distribution over count probabilities

$$p(\Pi|X, Z) = \int p(\Pi|F) \, p(F|X, Z) \, \mathrm{d}F \qquad (52)$$

contains the Gaussian process prior $p(F|X, Z)$ over $F$. The mapping from $F$ to $\Pi$ denoted by $\Pi(F)$ (Equation 1) is deterministic, and therefore $p(\Pi|F)$ is a delta distribution $\delta(\Pi - \Pi(F))$.

The exact Bayesian posterior over $\Pi$ and $Z$ is intractable, hence we use an approximate posterior as defined in Equation 3. The variational parameters $\varphi$ specify the latent variational posterior, while $\chi$ consists of inducing point locations $X_u$ and the means and covariance matrices of $q(U)$ for the

sparse Gaussian process posterior $q(F|X, Z)$ (Equation 46). The wrapped normal distribution used for circular dimensions in $q(Z)$, i.e. dimensions with $z \in [0, 2\pi)$, takes the form [43]

$$\mathcal{N}_{\text{wrap}}(z|\mu, \sigma^2) = \sum_{k=-\infty}^{\infty} \mathcal{N}(z|\mu + 2\pi k, \sigma^2) \tag{53}$$

and was evaluated with a finite cutoff at $k = \pm 5$ of the infinite sum. This is an accurate approximation as long as $\sigma \ll 2\pi$. When plotting the standard deviations of the approximate posterior $q(Z)$, we plot $\sigma$ for both Euclidean as well as circular variables. This is similarly an accurate approximation in the circular case when $\sigma \ll 2\pi$, which was true in practice.

The marginal likelihood in Equation 51 is intractable. Instead, we minimize the negative ELBO or variational free energy loss objective using our approximate posterior

$$\mathcal{F}_{\theta,\chi,\varphi} = -\mathbb{E}_{Z \sim q_\varphi(Z)} \mathbb{E}_{\Pi \sim q_{\theta,\chi}(\Pi|X,Z)} \left[ \log \frac{P(Y|\Pi) \, p_\theta(\Pi|X,Z) \, p_\theta(Z)}{q_{\theta,\chi}(\Pi|X,Z) \, q_\varphi(Z)} \right] \tag{54}$$
$$= \mathcal{F}_{\text{lik}} + \mathcal{F}_{\text{reg}}$$

which is an upper bound to the negative log marginal likelihood [32, 41]. The objective decomposes into a log likelihood expectation term $\mathcal{F}_{\text{lik}}$ and some regularization terms arising from the model priors $\mathcal{F}_{\text{reg}}$. These terms are amenable to Monte Carlo evaluation or quadrature approximation as we show next, and in some cases are even available in closed form.

The variational expectation of the log likelihood

$$\mathcal{F}_{\text{lik}} = -\mathbb{E}_{Z \sim q_\varphi(Z)} \mathbb{E}_{\Pi \sim q_{\theta,\chi}(\Pi|X,Z)}[P(Y|\Pi)] \tag{55}$$

can be evaluated using Monte Carlo sampling to obtain unbiased estimates in the general case. As an alternative method, Gauss-Hermite quadratures can provide a deterministic approximation to the expectation with respect to $q(F|X, Z)$ [41]

$$\mathbb{E}_{\Pi \sim q_{\theta,\chi}(\Pi|X,Z)}[P(Y|\Pi)] = \mathbb{E}_{F \sim q_{\theta,\chi}(F|X,Z)}[P(Y|\Pi(F))] \tag{56}$$

where $\Pi(F)$ denotes the transformation from $F$ to count probabilities as in Equation 1. Here we used

$$q(\Pi|X, Z) = \int \delta(\Pi - \Pi(F)) \, q(F|X, Z) \, \mathrm{d}F \tag{57}$$

analogous to Equation 52 for the generative model. This corresponds to a zero variance estimator with a small bias for sufficiently many quadrature points. As the likelihood factorizes over time, the expectation with respect to the multivariate variational posterior $q(F)$ factorizes into expectation terms with univariate Gaussian distributions, their variances taken from the diagonal of the covariance matrix. Because of this, we only need MC sampling from univariate distributions, allowing us to work with many time points and large batch sizes. This removes correlations between posterior function sample points at different input values, which cannot be done when factorization over time does not hold (e.g. GP priors in deep GPs). The issue of efficiently sampling from the full posterior has been considered in [39].

The regularization terms can be written as Kullback-Leibler divergences

$$\mathcal{F}_{\text{reg}} = D_{\text{KL}}(q_{\theta,\chi}(\Pi|X,Z)||p_\theta(\Pi|X,Z)) + D_{\text{KL}}(q_\varphi(Z)||p_\theta(Z)) \tag{58}$$

with the ratio of $q_\theta(\Pi|X, Z)$ and $p_{\theta,\chi}(\Pi|X, Z)$ in the first KL divergence equivalent to

$$D_{\text{KL}}(q(\Pi)||p(\Pi)) = \int \delta(\Pi - \Pi(F)) \, q(F) \, \log \frac{q(\Pi)}{p(\Pi)} \, \mathrm{d}F \, \mathrm{d}\Pi = \mathbb{E}_{q(F)} \left[ \log \frac{q(\Pi(F))}{p(\Pi(F))} \right] \tag{59}$$

where we have made the mapping $\Pi(F)$ explicit. In practice, the choice of $C < K$ implies this mapping is underparameterized and generally injective for matrices $W$ of rank $\geq C$. When we are in the universal limit $C = K$ and $W$ is rank $K$, the mapping will be bijective. As long as the mapping from $F$ to $\Pi$ is not many-to-one, we have the continuous random variable transform

$$p(\Pi) \, \mathrm{d}\Pi = p(F) \, \mathrm{d}F \tag{60}$$

and this leads to Equation 59 becoming $D_{\text{KL}}(q(F)||p(F))$. Hence, the regularization terms in the loss objective Equation 54 consist of KL divergences

$$\mathcal{F}_{\text{reg}} = D_{\text{KL}}(q_{\theta,\chi}(F|X,Z)||p_\theta(F|X,Z)) + D_{\text{KL}}(q_\varphi(Z)||p_\theta(Z)) \tag{61}$$

that can be computed with analytical expressions in the case when all distributions are Gaussian.

## E.2 Latent space priors

We use the Markovian priors as specified in Equation 2, and these priors can be specified on different manifolds [24]. For Euclidean spaces, we use the linear dynamical system prior

$$p(\boldsymbol{z}_{t+1}|\boldsymbol{z}_t) = \mathcal{N}(A\boldsymbol{z}_t, \Sigma) \tag{62}$$

In particular, we use diagonal $\Sigma$ and $A$ to learn factorized latent states. We constrain $A_{ii} = a_i \in (-1, 1)$ for stability, and we fix $\Sigma_{ii} = \sigma_i^2 = 1/(1 - a_i^2)$ to obtain a prior process with stationary variance 1 while optimizing for $a_i$. On the toroidal manifold, we use

$$p(\boldsymbol{z}_{t+1}|\boldsymbol{z}_t) = \mathcal{N}(\boldsymbol{z}_t + \boldsymbol{c}, \Sigma) \tag{63}$$

as due to rotational symmetry $A = I$. Again, we use diagonal $\Sigma$. Both $\boldsymbol{c}$ and $\Sigma_{ii} = \sigma_i^2$ are learned as part of the generative model.

When temporally batching input, one has to be careful to retain the continuity in the prior $p(Z)$ with the previous batch (beyond the first batch at the start). This is done by ensuring that the first $\boldsymbol{z}_t$ in the batch is the last step in the previous batch, and this will correctly subsample the prior $p(Z)$ defined over the entire input time series. When performing cross-validation with validation segments within the overall input time series, we treat the gap as a discontinuity in the latent trajectory and do not include the latent state right before the validation segment.

## E.3 Gaussian process kernel functions

In this work, we used the RBF kernel defined on Euclidean and toroidal manifolds [24]. In particular, this kernel function is given by

$$k(\boldsymbol{x}, \boldsymbol{y}) = \sigma^2 \, e^{-\frac{1}{2} \sum_{i=1}^{D} d_i^2(x_i, y_i; l_i)} \tag{64}$$

with rescaled distances

$$
\begin{aligned}
d_{\mathbb{R}}^2(x, y; l) &= \left( \frac{x - y}{l} \right)^2 \\
d_{\mathbb{T}}^2(x, y; l) &= 2 \left( \frac{1 - \cos(x - y)}{l} \right)^2
\end{aligned}
\tag{65}
$$

for Euclidean and toroidal spaces $\mathbb{R}$ and $\mathbb{T}$, respectively. To cover different input dimensions of different topologies, we use product kernels with suitable distances $d$ per input dimension, resulting in sums over dimensions in Equation 64. These distance functions can be used to extend other kernels such as Matérn kernels to non-Euclidean spaces [24].

## E.4 Overall algorithm and code

The outline of the inference procedure is given in Algorithm 1. Additionally, instead of drawing Monte Carlo samples for the Gaussian variational posterior $q(F)$, we provide the option to compute the Gaussian expectation using Gauss-Hermite quadratures [41]. This was used to estimate the cvLLs (Equation 9) for models after training, which reduced stochasticity in the cvLL estimate with a negligible bias when using 100 quadrature points. Monte Carlo samples or quadrature point dimensions are parallelized over in addition to other dimensions like neurons or time, using extra tensor dimensions in modern automatic differentiation libraries. We use PyTorch [44] to implement the algorithm for inference of our models. For optimization, we use Adam [45] with no weight decay and default optimizer hyperparameters in PyTorch.

The code provided[2] contains a library under the name 'neuroprob', which was written to organize the implementations of Gaussian process and GLM based models with different likelihoods as used for baseline models in this paper. In addition to count likelihoods, it contains an implementation of spike-spike and spike-history couplings [12, 46] and modulated renewal processes [47, 48] to deal with data at the individual spike time level. All models can be run with both observed and latent inputs on Euclidean and toroidal manifolds [24].

---

[2]`https://github.com/davindicode/universal_count_model`

---

**Algorithm 1** Joint latent-observed input inference scheme

---

**Input** spike counts $Y$, observed covariates (e.g. behaviour) $X$

1: Batch data over time dimension, taking into account continuity in $p(Z)$ (subsection E.2)
2: **while** $\mathcal{F}_{\theta,\chi,\varphi}$ not converged **and** iterations $<$ maximum **do**
3:     **for** each batch **do**
4:         Generate $m$ MC samples $z \sim q_\varphi(Z)$ (if relevant), $m$ copies of $x = X$
5:         Compute posterior $q(F|X, Z)$ over $C$ functions per neuron
6:         Generate $k$ MC samples of $F$ per input sample $(x, z)$
7:         Order samples of $F$ into vectors $\boldsymbol{f} \in \mathbb{R}^C$ for each neuron
8:         Evaluate basis expansion $\boldsymbol{a} = \boldsymbol{\phi}(\boldsymbol{f}) \in \mathbb{R}^{K+1}$
9:         Compute $P(y|\boldsymbol{a}) = \text{softmax}(W\boldsymbol{a} + \boldsymbol{b})$ for all neurons and time steps
10:        Compute loss $\mathcal{F}_{\theta,\chi,\varphi}$ (Equation 54)
11:        Compute gradients for parameters $\theta$, $\chi$ and $\varphi$
12:        Update parameters with a gradient step
13:     **end for**
14:     Adapt learning rates (if annealing)
15: **end while**

---

## E.5  Model fitting

### E.5.1  Inducing point initialization

The first input dimension had its inducing points uniformly spaced between $0$ and $2\pi$ for circular dimensions, and $-1$ to $1$ for Euclidean latent dimensions. Observed dimensions had natural intervals defined by the behavioural statistics (e.g. $0$ to the mean animal speed), and we placed inducing points uniformly throughout this interval. For the other dimensions, we initialized random inducing point locations based on the topology of the input variable. We place Euclidean variables as a random uniform distribution in its corresponding interval as described previously, while circular variables took on random uniform values in $[0, 2\pi]$.

The number of inducing points has been shown to scale favourably as $O((\log T)^D)$ for standard Gaussian process regression models [49]. In this work, we used $O(D \log T)$ which captured rich tuning and satisfactory model fits combined with the flexible count distributions. The suggested $O((\log T)^D)$ does become computationally expensive for high dimensional input, and was not tried with the high-dimensional regression models.

### E.5.2  Fitting details

We select the model with the lowest loss from 3 separate model fits, initialized with randomized inducing points as described above. The maximum number of training epochs was 3000, but we stopped training before if the loss did not decrease more than $\approx 10^{-3}$ percent over 100 steps. The learning rate was set to $10^{-2}$, and we also anneal the learning rate every 100 steps by a factor 0.9. This was for both Gaussian process as well as artificial neural network models. In the case of latent spaces, we used a learning rate of $10^{-3}$ for standard deviations of the variational distribution $q(Z)$. All cases lead to satisfactory convergence of the model.

For latent variable models with a single angular latent, we initialize the lengthscale at large values. This avoided the model to overfit and fold the latent space as seen in panel A of Figure 3 for the ANN model. For these models, the best fits were achieved with an initial learning rate of $3 \cdot 10^{-2}$ and $5 \cdot 10^{-3}$ for the kernel lengthscale and the standard deviations of the variational distribution $q(Z)$.

### E.5.3  Hardware and fitting time

Synthetic data was analyzed with GeForce RTX 2070 (8 GB of memory). Real data was analyzed with Nvidia GeForce RTX 2080Ti GPUs (with 11 GB of memory). Fitting 33 neurons with $\sim 6 \cdot 10^4$ time points with the regression model in Figure 3 takes around 20 minutes, while fitting with a four dimensional latent spaces added takes around 50 minutes. These numbers can fluctuate depending on the flexible stopping criterion above. Generally, there is a trade-off between memory usage and speed by setting the batch size, with larger batch sizes being generally faster but taking more memory.