# OpenReview forum: "A universal probabilistic spike count model reveals ongoing modulation of neural variability"
_NeurIPS.cc/2021/Conference — NeurIPS 2021 Poster_

### Official Review · Reviewer_Saxk · 2021-07-16

**Rating:** 6
**Confidence:** 2

**Summary:**

In this paper authors provide a method for modelling spike counts as a function of input covariates and latent covariates. A Markovin prior is assumed over latent parameters, and the effect of latent parameters and input covariates on spike counts are modelled using a Gaussian process. The authors also suggest a method for calculating Z-scores and tuning index based on their method. The results include analyses of a synthetic dataset and an experimental data, which generally shows the proposed method outperforms considered baselines.

**Ethical Concerns:**

Please include relevant ethics information regarding the experimental data.

**Limitations And Societal Impact:**

They are adequately discussed.

**Main Review:**

The paper is generally clear (but see below). The statistical model is interesting and suitable for modelling the joint effect of latent and observed covariates on the neural activities. (the main text needs to be updated based on the typo mentioned in the supplementary materials). The results are also encouraging and show that the model is improving on baseline methods.

I have the following concerns about the study that require further improvements:

Some of the results are presented in a rather confusing way. For example cvLL results are presented relative to the Poisson model in some panels (e.g,, Fig 2A) but not in other places (e.g., Fig 3 in supple). In Fig 2A for example it's hard to see whether differences with the performance of the Poisson model is significant or not. Also RMSE is reported for synthetic experiments, but not for the experimental data.

The baselines are also different between the experiments; the performance of ANN is not reported for the experimental data. Also, I couldn’t find how the architecture of the neural networks was selected; how many neurons? How were hyper-parameters tuned? This is particularly important since recent approaches report promising results using recurrent neural networks, e.g. below, and the reflective performance of the suggested method w.r.t.to these methods is unclear.

Pandarinath, Chethan, et al. "Inferring single-trial neural population dynamics using sequential auto-encoders." Nature methods 15.10 (2018): 805-815.

Sections 2.4 and 2.5 are rather hard to follow. For example it would be good to expand why the definition of Z in eq 8 is a sensible one and how it is related to the common definition of z score. Below eq 11, for example it's unclear what does data being more/less regular than the data mean.


**Time Spent Reviewing:**

3

---

> ### Author Response · Authors · 2021-08-09
> **Response to Reviewer Saxk**
>
> We thank the Reviewer for their constructive feedback. We provide a detailed point-by-point response to all their comments below.
>
> **The paper is generally clear (but see below). The statistical model is interesting and suitable for modelling the joint effect of latent and observed covariates on the neural activities.**
>
> Thank you.
>
> **(the main text needs to be updated based on the typo mentioned in the supplementary materials).**
>
> We will update the paper by fixing the typo mentioned in the Supplementary Materials.
>
> **The results are also encouraging and show that the model is improving on baseline methods.**
>
> Thank you.
>
> **I have the following concerns about the study that require further improvements:
> Some of the results are presented in a rather confusing way. For example cvLL results are presented relative to the Poisson model in some panels (e.g,, Fig 2A) but not in other places (e.g., Fig 3 in supple).**
>
> We will plot all cvLL comparisons (Fig.2A, Fig.3D) relative to the Poisson baseline, as is currently done in Fig.2A (with $\Delta$cvLL).
>
> **In Fig 2A for example it's hard to see whether differences with the performance of the Poisson model is significant or not.**
>
> The universal model consistently improves over baselines in the cvLL (p=0.18 for Fig.2A, p=0.08 for Fig.3D, one sample t-test with respect to Poisson baseline, although with only a few data points i.e. cross validation folds, we will increase the number of data points and report the p-values in the paper). We want to emphasise that in the study of neural variability it is not mainly the “performance” of these models as measured in cvLL that matters, but the extent to which the model captures important, neurobiologically relevant modulations of response variability (see Introduction). In this regard our model clearly outperforms the baseline models that fail to capture count statistics sufficiently, see the Kolmogorov-Smirnov statistic $T_{KS}$ in the top right panel of Fig. 2A where we see significant improvements for many individual cells. From the results, we discover that the data can be both under- and overdispersed albeit in different regimes, depending on the mean firing rate and various covariates (as shown e.g. in Fig. 2B-C). Therefore the difference from Poisson-like statistics is not just simply overdispersed or underdispersed, and our method has the capacity to reveal these subtle differences.
>
> **Also RMSE is reported for synthetic experiments, but not for the experimental data.**
>
> In Fig.1A, RMSE was computed for inferred latents wrt their true values used to generate the synthetic data. In general, this is obviously not possible for real data (Fig.2), as we do not have access to the true value of latent variables (otherwise, they would not be latent). Although Fig.3D also presentes the analyses of real data, here we did not even include head direction (the most important determinant of head-direction cell firing) as an observed covariates, and thus we could make the educated guess that the single latent variable that our method extracted must correspond to the true value of this variabile (which in reality is observed in the experiment, we just did not allow our algorithm to access this information). This was motivated by the success of previous approaches to the unsupervised analysis of head direction cell data (see Ref. [38] in main text or [34] in supplementary). Therefore, we were able to compute the RMSE in this exceptional case. We will expand on the current explanation of this in the supplementary (Section A.3)
>
> **The baselines are also different between the experiments; the performance of ANN is not reported for the experimental data.**
>
> Two out of the three controls in the synthetic experiment (Fig.1), we also use for the experimental data analysis (Fig.2): Poisson, NBh. These are important baselines in the sense that they are fundamentally different from our model and there have been previous influential proposals that neural variability is well captured by these models (Refs. [46], [50], [55] in main text). The purpose of including the U(ANN) in Fig.1 was not to provide such a baseline but instead to test different model components in the process of developing the final version of our model. Note that, as normal while developing such models, there have been many such “component tests” that we don’t even show.  Once we decided on the best combination of components of our model based on the validation tests, we used that form throughout the rest of the paper. We will explain this better when we describe the results with U(ANN).
>
> **Also, I couldn’t find how the architecture of the neural networks was selected; how many neurons? How were hyper-parameters tuned**
>
> We apologise for the lack of these details, we will include these in the final version of the paper (Supplementary Material). Briefly, we used a feedforward neural network that was sufficiently expressive to capture the covariate dependence of spike count distributions. Our ANN consisted of 3 fully connected hidden layers of 50, 50 and 100 units with sinusoidal activation functions to encourage smoother outputs (Sitzmann et al. `Implicit Neural Representations with Periodic Activation Functions’, NeurIPS 2020). The input layer provided observed X and latent Z covariates, with angular variables expanded as 2D input (cos(x),sin(x)). The output layer consisted of $N \cdot C$ linear units providing $f_{cn}$ in Eq.1, with $N$ the number of neurons and $C=3$ the number of degrees of freedom per neuron. Learning rates were identical to those used in the Gaussian process model.
>
> **This is particularly important since recent approaches report promising results using recurrent neural networks, e.g. below, and the reflective performance of the suggested method w.r.t.to these methods is unclear.
> Pandarinath, Chethan, et al. "Inferring single-trial neural population dynamics using sequential auto-encoders." Nature methods 15.10 (2018): 805-815.**
>
> We note that the goal of the ANN used within our U(ANN) is fundamentally different from the way ANNs have been used in previous approaches to modelling neural data, including LFADS. For us, the ANN is simply used as an *instantaneous* mapping from covariates to spike count distributions (hence we used a feedforward network, see above). Thus, recurrent neural networks were not appropriate for our purposes. In general, ANNs (feedforward or recurrent) suffer from the lack proper regularization in regimes where data is lacking, while GPs remain well calibrated -- we will explain this better in the final paper where we describe the U(ANN) vs. U(GP) comparison.
>
> More broadly, while the LFADS approach can be very powerful, it is not ideal for our use-domain: it generally requires very large amounts of trial-structured data, and it does not easily allow the flexible modeling of modulations of neural variability. We will cite LFADS and include this point briefly in the Discussion.
>
> **Sections 2.4 and 2.5 are rather hard to follow. For example it would be good to expand why the definition of Z in eq 8 is a sensible one and how it is related to the common definition of z score.**
>
> We explain the relation between the commonly used definition of Z-score and our generalization in the Supplementary Material (Section C.1), but we will make it more explicit that our definition subsumes the classical one while remaining unbiased even at low spike counts under the special case of a Poisson model, and most importantly is agnostic to the underlying count distribution (necessary as our method can model arbitrary spike count distributions). We will also mention this in Section 2.4 of the main text.
>
> **Below eq 11, for example it's unclear what does data being more/less regular than the data mean.**
>
> We apologise for the confusion, we simply meant under- or overdispersed (concepts that we thoroughly introduce and discuss in the paper). We will rewrite this in the paper. In general, we will go through Section 2.4-5 with a fresh pair of eyes to make sure our explanations are clear.

---

### Official Review · Reviewer_XYHL · 2021-07-16

**Rating:** 7
**Confidence:** 3

**Summary:**

The paper presents a model of spike count distributions in a population of neurons. The model’s spike count distribution depends both on stimulus information and latent variables through a Gaussian process to capture a large range of joint spike count distributions (as the number of factors in the model increases). Moreover, the authors present a set of model fit statistics for evaluating the model fit, and how correlations factor into the model’s predictions.

**Limitations And Societal Impact:**

Yes

**Main Review:**

The model improves upon previous non-parametric approaches (such as the universal binary count model) to modeling spike count distributions in neural population recordings: importantly, the proposed model includes stimulus information and temporal correlations. The example applications are very thorough in examining the model fit and comparisons to the Poisson and negative binomial models.

The presentation was generally clear. However, I am not convinced I understand the procedure for quantifying correlations given on lines 183-190. I’m certain I’m overthinking things here, but I do think this one portion of the methods could be presented more clearly.

Minor point: one limiting factor not mentioned in the discussion is that the stimulus term in the examples here are low-dimensional compared to stimuli in sensory systems.

**Time Spent Reviewing:**

2.5

---

> ### Author Response · Authors · 2021-08-09
> **Response to Reviewer XYHL**
>
> We thank the Reviewer for their constructive feedback. We provide a detailed point-by-point response to all their comments below.
>
> **The model improves upon previous non-parametric approaches (such as the universal binary count model) to modeling spike count distributions in neural population recordings: importantly, the proposed model includes stimulus information and temporal correlations. The example applications are very thorough in examining the model fit and comparisons to the Poisson and negative binomial models.**
>
> Thank you.
>
> **The presentation was generally clear. However, I am not convinced I understand the procedure for quantifying correlations given on lines 183-190. I’m certain I’m overthinking things here, but I do think this one portion of the methods could be presented more clearly.**
>
> We apologize if our explanation was not sufficiently clear. We simply referred to the universal fact that, in principle, the observed correlation between two random variables can always be captured by a suitable choice of a latent variable *z*, conditioned on which the two observed variables are independent. Whether the latent variable (its distribution, and the way observed variables depend on it) is suitable in this sense can be judged post hoc by testing whether this conditional independence holds empirically.
> We follow this logic by focussing on pairwise second-order correlations (as testing general independence in high dimensional data is not practically feasible). We simply use the Pearson correlation to quantify neural correlations, as standard in the field (Ref. [17]). We first measure correlations in the posterior predictive distribution of the model *without* latent variables. This is our estimate of “raw” noise correlations in the data. (We use the model’s posterior predictive in order to efficiently factor out the effect of observed covariates *X*, and thus isolate noise correlations from raw total correlation in the data that would otherwise also include signal correlations.) We then compute the same correlations under the posterior predictive distribution of the model that does include latent variables -- this is what Eq.13 shows. This is our model of residual correlations after conditioning on *z*. For each set of correlations (with and without latents), we use an appropriate statistical test (using Fisher’s Z-transform) to see if they are significantly different from zero. Our main finding is that both in validation (Fig.1) and in real data (Fig.2), raw noise correlations can be large (significantly different from zero), while they become not (significantly) different from zero after conditioning on *z* -- suggesting that the low dimensional latent variables we introduced and inferred were suitable for explaining away neural correlations.
>
> We will improve our explanation in the paper along these lines.
>
> **Minor point: one limiting factor not mentioned in the discussion is that the stimulus term in the examples here are low-dimensional compared to stimuli in sensory systems.**
>
> This is indeed an important point. One could approach high dimensional input using deep kernels (Wilson et al. `Deep Kernel Learning', AISTATS 2016), which combines ANN with covariance kernels to handle high dimensional input such as images while retaining benefits from kernel approaches. We will add this discussion point to the Discussion.

---

> > ### Comment · Reviewer_XYHL · 2021-08-27
> > **Reply**
> >
> > I thank the authors for their detailed response. They have answered all of my questions.

---

### Official Review · Reviewer_S8Uc · 2021-07-16

**Rating:** 6
**Confidence:** 3

**Summary:**

The authors present a very broad framework to model multi-neuronal spike trains (multivariate time series over the nonnegative integers). The model class is sufficiently broad that it can express any arbitrary distribution over N-dimensional vectors containing the integers {0, ..., K}, and thus model the activity of N neurons within a single time bin (assuming an upper bound on the number of spikes, K).

**Limitations And Societal Impact:**

The limitations of the model are clear and I do not foresee any negative societal impacts.

**Main Review:**

This paper is well-written, the figures look polished, and I believe the core claims of the paper are correct. Overall, I would like to see this paper published if the authors can address my concerns below, which mostly deal with the interpretation of the model, relation to existing methods, and framing of the paper. In short, I think the model could be useful in cases where one would like to specify large time bin sizes, but the utility of the method is less clear in the common case where one takes the time bins to be very short.

Most existing methods, the authors claim, have a "fundamental underlying assumption of Poisson spiking statistics." The model they propose does not have this assumption if time bins are large and K > 1. However, if we consider the limit of very small time bins (on the order of 1 millisecond) then it is reasonable to set K=1, since the probability of a neuron firing more than one spike is negligible. The multinomial distribution in thier model becomes a binomial distribution. In the limiting case of of time bin duration going to zero, you recover Poisson statistics (see, e.g., sec 5.2 of Kass et al. "Analysis of Neural Data"). In this limit, the model proposed by the authors looks very similar to existing models, like GPFA or GLMs.

If the above analysis is correct, then I think the authors should acknowledge it in the paper and explain that their method is a generalization that retains "universal" modeling capabilities when time bins are not very small and when K > 1. Then, they should explain what the relative advantages and disadvantages are to choosing larger time bins.

There are other parts of this paper that I really like, such as the combination of supervised learning and latent variable modeling to capture representational drift and head direction tuning. However, these modeling innovations don't seem to be sold as the key results of the paper and it doesn't seem like the "universal" spike count model is really necessary to enable this analysis. Again, it seems like one could choose small time bins and use GPFA (w/ poisson noise) + GLMs to arrive at a similar result.

Overall, I thought this paper was very well-written and presented some elegant analysis of real experimental data, so I have scored it as a marginal accept. The modeling innovations that are most heavily emphasized in the manuscript seem to be dependent on having large time bin sizes. If my interpretation here is wrong I am happy to raise my score higher.

**Time Spent Reviewing:**

4

---

> ### Author Response · Authors · 2021-08-09
> **Response to Reviewer S8Uc**
>
> We thank the Reviewer for their constructive feedback. We provide a detailed point-by-point response to all their comments below.
>
> **This paper is well-written, the figures look polished, and I believe the core claims of the paper are correct.**
>
> Thank you.
>
> **Overall, I would like to see this paper published if the authors can address my concerns below, which mostly deal with the interpretation of the model, relation to existing methods, and framing of the paper. In short, I think the model could be useful in cases where one would like to specify large time bin sizes, but the utility of the method is less clear in the common case where one takes the time bins to be very short.
> Most existing methods, the authors claim, have a "fundamental underlying assumption of Poisson spiking statistics." The model they propose does not have this assumption if time bins are large and K > 1. However, if we consider the limit of very small time bins (on the order of 1 millisecond) then it is reasonable to set K=1, since the probability of a neuron firing more than one spike is negligible. The multinomial distribution in their model becomes a binomial distribution. In the limiting case of time bin duration going to zero, you recover Poisson statistics (see, e.g., sec 5.2 of Kass et al. "Analysis of Neural Data"). In this limit, the model proposed by the authors looks very similar to existing models, like GPFA or GLMs. If the above analysis is correct, then I think the authors should acknowledge it in the paper and explain that their method is a generalization that retains "universal" modeling capabilities when time bins are not very small and when K > 1.**
>
> The Reviewer is correct in that in the limit of very small time bins and K=1 our model becomes a generalisation of the universal binary model (Ref. [33]), allowing (for observed and latent) covariates -- which is critical for dissecting signal and noise correlations in neural data. In fact at K=1, the distinction between Poisson and non-Poisson is irrelevant as all possible spike count distributions in a time bin are Bernoulli. As the Reviewer points out, in this limit our model also becomes conceptually very similar to GLMs and GPFAs (with binary emissions) in that conditioned on covariates (potentially including the spiking history of a neuron itself, or of other neurons, as in GLMs) spiking becomes an inhomogeneous Poisson process. However, these previous models remain Poisson-like *at all time scales* (in the sense that, conditioned on covariates, spike counts remain Poisson distributed) and only allow covariates to modulate the mean firing rate. (This is because they implicitly assume spiking in consecutive infinitesimal time bins to be independent given covariates). The key advance of our work is precisely that *even at larger time bins* (and with K>1) it is not restricted to Poisson spike count distributions and allows covariates to modulate any spike count statistic. Indeed, we found that experimental data deviated from Poisson-like statistics in important ways and was in many cases substantially modulated by covariates at the 40 ms time bin size (and K=11) we chose (Figs. 2B and 2C). Our choice of 40 ms time bins was based on previous empirical measures of autocorrelation time scales in neural activity (e.g. Azouz and Gray, J Neurosci 1999; Berkes et al Science 2011). (We also made sure none of our behavioural covariates had a shorter time scale, see Fig.2G.) In follow-up work, we are exploring model-selection based methods to choose this time bin automatically. We will discuss these issues more thoroughly in the final version of the paper (Discussion, section on Related work, with more technical parts of the arguments above in the Supplementary Material, see also below).
>
> **Then, they should explain what the relative advantages and disadvantages are to choosing larger time bins.**
>
> The advantage of choosing essentially infinitesimally small time bins is that there is no “arbitrary” (though see above) time bin size parameter and every individual spike can be predicted (at least in principle, see also note below). The disadvantage is that, as we explain above, in this case it is not even conceptually possible to model modulations of response variability independent from modulations of firing rates, as the two are inseparable in the underlying Bernoulli model (as K=1). Indeed, experimental studies of neural variability, and its modulation by covariates, have always used larger time bins to compute Fano factors or Z-scores (Refs. [10], [8], [15], [12]). Our work is a direct generalization of this perspective, extending beyond rigid trial-structure. At the other extreme, the disadvantage of choosing time bins that are too large is that it may miss the time scale at which covariates modulate neural firing. This is why we made sure our time bin size was still smaller than the time scale at which our covariates themselves varied (see above). We will include a reference line at our time bin width in Fig.2G (lying below all data points shown, at $\approx$ -1.4) to emphasise this. We will include a separate section on the choice of time bin width, including these arguments, in the Supplementary Material.
>
> We also note that time-rescaled renewal (and more general point) process models (Refs. [45], [77], [83], [84]) might seem as a viable alternative to our approach as they allow predicting individual spikes without making an underlying independent Bernoulli assumption and thus without restricting to conditioned spike count statistics to be Poisson. Indeed, we did consider such an approach. However, these models are difficult to generalise such that they allow true heteroscedasticity in the way our model does (i.e. independent modulations of mean and variability). We will also mention this in the “Related work” section of the Discussion.
>
> **There are other parts of this paper that I really like, such as the combination of supervised learning and latent variable modeling to capture representational drift and head direction tuning. However, these modeling innovations don't seem to be sold as the key results of the paper and it doesn't seem like the "universal" spike count model is really necessary to enable this analysis. Again, it seems like one could choose small time bins and use GPFA (w/ poisson noise) + GLMs to arrive at a similar result.**
>
> The Reviewer is correct that these contributions are largely independent of our new method -- we also indicate this by prefacing the corresponding paragraph by saying that “Beside our main contribution…” (line 268) and by having relegated the figure showing these results (Fig.3C) into the Appendix. Nevertheless, we did want to at least mention these results in the main text as the presence of a systematic representational drift is a novel finding (for head direction cells) with potential biological relevance that we noted during our analysis. The methodological innovation here was to use GP-regression with time as a regressor to study representational drift, but -- as the Reviewer correctly points out not -- indeed not specifically the application of the universal spike count model per se. We will make it even more explicit that these results do not necessarily depend on using the universal spike count model.
>
> **Overall, I thought this paper was very well-written and presented some elegant analysis of real experimental data, so I have scored it as a marginal accept. The modeling innovations that are most heavily emphasized in the manuscript seem to be dependent on having large time bin sizes. If my interpretation here is wrong I am happy to raise my score higher.**
>
> The Reviewer’s interpretation and comments are technically correct, but to reiterate: there is nothing to say a priori that *the right* time bin size for studying neural variability is the infinitesimally small limit used by several previous approaches. Indeed, our empirical results showing non-Poisson conditioned spike count distributions at longer time scales suggest that longer time scales may be more appropriate -- at least in the data set we analysed. In general, we argue (see above) that if one wants to study the modulation of neural response variability then one must use appropriately sized (non-infinitesimally small) time bins. In turn, in this setting, our approach is unique in offering a statistically principled method to do so and offers novel insights into the variability of head direction cells. As we noted in our previous responses, we will explain these points about the pros and cons of choosing different time bin sizes for analysis, and the fact that there is not one universally best choice, more clearly in the final paper.

---

### Official Review · Reviewer_oo8R · 2021-07-30

**Rating:** 6
**Confidence:** 2

**Summary:**

This study proposes a statistical model to jointly estimate the mean firing rate and covariance structure based on the spike count of neural populations, and test the method on mouse head-direction cells.

**Limitations And Societal Impact:**

Yes

**Main Review:**

The paper proposes a new descriptive statistical model to fit neural data. By using Gaussian process, the model is able to jointly fit mean firing rate (1st order statistics) and covariance (2nd order statistics) simultaneously, which is  the main novelty of this paper. And the author used this model to fit synthetic data and real neural data, and compared the model performance with other alternative models used in neural data analysis. The study is solid, and figures look clean and great. However, since some key math notations are not defined yet, I am not sure the math used in this study is correct or not although I hope so.

### Writing:
The writing and the definition of math notations can be improved a lot in a revised version. For example, some underlying motivations of proposing a math equation are lacking, and some math notations are not defined at all when using.

- I don’t understand what are observed X and latent Z, and what is the difference between the two variables. From the generative model shown in Fig. 1, the observed X seems to be a latent variable generating recorded spike count Y. Or X is regarded as the presented stimulus to an animal? Please give a concrete example of what X and Z can be referred to in neurophysiology experiments and otherwise it is not easy to understand.
is the motivation to include those two variables.

- Line 100: It is not unclear what $\Pi$ is. It is not defined. The $\Pi$ seems a lumped variable including all observations but I am not sure.
- Eq.8: what is \phi^{-1}? it is not defined.
- Eq.11: it is not clear what F(q) is. What is the difference between $F(q)$ and $q$? Is $F(q)$ the empirical distribution while $q$ the parametric distribution defined in Eq. 2?
- Eq. 12: please add the motivation of including $1/T+ 1/3T^2$ in the equation.


**Time Spent Reviewing:**

2.5

---

> ### Author Response · Authors · 2021-08-09
> **Response to Reviewer oo8R**
>
> We thank the Reviewer for their constructive feedback. We provide a detailed point-by-point response to all their comments below.
>
> **The paper proposes a new descriptive statistical model to fit neural data. By using Gaussian process, the model is able to jointly fit mean firing rate (1st order statistics) and covariance (2nd order statistics) simultaneously, which is the main novelty of this paper. And the author used this model to fit synthetic data and real neural data, and compared the model performance with other alternative models used in neural data analysis. The study is solid, and figures look clean and great.**
>
> Thank you. Just to be precise: it is not just 1st and 2nd-order spike count statistics, but in principle statistics of *any* order that our model allows covariates to modulate as we infer full spike count distributions. It is true, however, that *post-hoc*, after fitting the model, we focus on how the 1st and 2nd-order statistics of the inferred spike count distributions are modulated because this is what we believe we can reliably estimate given the amount of data, and this is what has been studied previously with statistically more naive approaches (representing the basis of comparison for our results).
>
> **However, since some key math notations are not defined yet, I am not sure the math used in this study is correct or not although I hope so.
> The writing and the definition of math notations can be improved a lot in a revised version. For example, some underlying motivations of proposing a math equation are lacking, and some math notations are not defined at all when using.**
>
> We apologise for the imperfections of the mathematical notation. As you can see from our supplementary notes on typos, we have already started correcting these. As a result, the revised version will contain cleaner and tidier notation. We are particularly grateful for your specific points of feedback, and will rewrite the relevant parts of the text.
>
> **I don’t understand what are observed X and latent Z, and what is the difference between the two variables. From the generative model shown in Fig. 1, the observed X seems to be a latent variable generating recorded spike count Y. Or X is regarded as the presented stimulus to an animal? Please give a concrete example of what X and Z can be referred to in neurophysiology experiments and otherwise it is not easy to understand. What is the motivation to include those two variables.**
>
> We apologies if this was not clear from the text as this is indeed fundamental to our analysis. In Fig.1, we used standard notation to denote observed variables with grey shading and latent variables with a white background. Fig.4 of the Supplementary Material was intended to further clarify this, although we did not have references to it in the main text (we will add these in the revision). In the text, we do say in general that X is observed and Z right upfront, where we define them (lines 95-97). We describe concrete examples when we describe the analysis of experimental data (lines 235-237):  “Regression was performed against head direction (HD), angular head velocity (AHV), animal speed and position, and absolute time, which collectively form $X_{\mathcal{D}}$ in this model.” (We realise that the subscript $\mathcal{D}$ may be confusing so we will omit this in the revision.) We will include such concrete examples right at the beginning when we introduce this notation. Similarly, we will say that Z could capture uncontrolled (low-dimensional) time-varying factors influencing neural responses, such as eg. the level of attention or uncertainty. In addition -- in line with previous approaches (Refs. [31] and [38]) --, Z can also be used to capture factors that are in reality controlled and/or observed but are not presented to the model as such (i.e. not included in X), to test if they can be recovered in an unsupervised way (see eg our analysis in Fig.3D extracting head direction using such an unsupervised approach).
>
> **Line 100: It is not unclear what is $\Pi$. It is not defined. The
>  seems a lumped variable including all observations but I am not sure.**
>
> $\Pi$ is defined somewhat implicitly and tersely in line 100 by saying what $p(\Pi | X)$ is: “the joint count distribution of a population”. We will expand on it to say that $\Pi$ represents all the spike count probabilities of a population, across time steps and neurons, which then can be used in the likelihood when fitting to spike count data. To be more specific, each neuron at each time step has spike count probabilities in a vector $\boldsymbol{\pi}$ with K+1 elements (0 to K counts), and then $\Pi$ describes all $\boldsymbol{\pi}_{nt}$ at all time steps $t$ and neurons $n$.
>
> **Eq.8: what is $\Phi^{-1}$? it is not defined.**
>
> Apologies, we thought $\Phi(x)$ was standard notation for the standard normal CDF (and $\Phi^{-1}$ for its inverse), but we will make this explicit in the text.
>
> **Eq.11: it is not clear what F(q) is. What is the difference between $F(q)$ and $q$? Is $F(q)$ the empirical distribution while the $q$ parametric distribution defined in Eq. 2?**
>
> $q$ in Eq.8 and 11 is the quantile value of a data point under the (fitted) model’s predictive distribution, expressing the probability with which the model predicted a data point that was less than or equal to that actually observed. As such, $q$ is between 0 and 1, and it should have a uniform distribution if the model was a perfect fit to the data. $F(q)$ is the empirical cumulative distribution function of $q$, expressing the fraction of data points that were associated with a quantile value less than or equal to $q$. Again, in the ideal case, this should be the identity function (the cumulative of the uniform distribution on the unit interval) -- this is precisely what the KS test we are using is testing in a statistically principled way. We will explain this better in the text (rather than just citing Ref. [45] for the well-versed reader).
>
> $q_{\varphi,\vartheta}$, $q_{\varphi}$, and $q_{\vartheta}$ in Eq. 2 are completely different objects (the variational posterior, and it components, used for inference). As such they are not related to $q$ above at all. We sincerely apologise for this clash of notation. We will use $u$ to refer to quantiles instead, and keep $q$ to refer to variational posterior and its components.
>
> **Eq. 12: please add the motivation of including $1/T+1/(3T^2)$ in the equation.**
>
> In appendix D (which we refer to in line 181 below Eq.12), we derive the asymptotic moments of the sampling distribution of the $T_{DS}$ statistics based on the sampling distribution of the logarithm of the chi-squared variable. This leads to this term being necessary to remove bias away from $0$ for $T_{DS}$ in the case when the model is correct (null hypothesis where $T_{DS}$ should be zero, i.e. the data is neither under- nor overdispersed with respect to the model). We will briefly mention this in the main text surrounding Eq.12.

---

### Decision · Program_Chairs · 2021-09-27

**Decision:**

Accept (Poster)

**Comment:**

This paper presents a universal spike-count model for flexibly describing both over-dispersion and under- dispersion in neural spiking data. The model is presented with an efficient variational fitting method, and can be applied to data without known stimuli, as is often the case for overdispersion metrics. The reviewers recognized the novelty of the model, and the topic fits nicely in the probabilistic modeling and neuroscience focuses in NeurIPS. Moreover spike-count models are an ongoing discussion in the computational neuroscience community (heavily represented at NeurIPS) and this work presents a new and interesting addition to the literature. I therefore recommend this work be accepted in NeurIPS.